# Tuning movement for sensing in an uncertain world

Chen Chen[1,2], Todd D Murphey[1,3], Malcolm A MacIver[1,2,3,4]*

[1]Center for Robotics and Biosystems, Northwestern University, Evanston, United States; [2]Department of Biomedical Engineering, Northwestern University, Evanston, United States; [3]Department of Mechanical Engineering, Northwestern University, Evanston, United States; [4]Department of Neurobiology, Northwestern University, Evanston, United States

**Abstract** While animals track or search for targets, sensory organs make small unexplained movements on top of the primary task-related motions. While multiple theories for these movements exist—in that they support infotaxis, gain adaptation, spectral whitening, and high-pass filtering—predicted trajectories show poor fit to measured trajectories. We propose a new theory for these movements called energy-constrained proportional betting, where the probability of moving to a location is proportional to an expectation of how informative it will be balanced against the movement's predicted energetic cost. Trajectories generated in this way show good agreement with measured trajectories of fish tracking an object using electrosense, a mammal and an insect localizing an odor source, and a moth tracking a flower using vision. Our theory unifies the metabolic cost of motion with information theory. It predicts sense organ movements in animals and can prescribe sensor motion for robots to enhance performance.

## Introduction

Movement can be used to obtain information that is unevenly distributed in the environment. Because movement is energetically costly, there is likely a balance between the benefits of increased sensory information and energetic costs for obtaining that information (*MacIver et al., 2010*). We have developed a theory that unifies these two dimensions of information acquisition and can be applied across sensory modalities and species. This theory, energy-constrained proportional betting, predicts the small and seemingly extraneous movements that sensory organs or animals undergo as they near or track a target of interest (see *Figure 1—figure supplement 1*; *Martin, 1965*; *Basil et al., 2000*; *Ferner and Weissburg, 2005*; *Webb et al., 2004*; *Willis and Avondet, 2005*; *Porter et al., 2007*; *Louis et al., 2008*; *Duistermars et al., 2009*; *Yovel et al., 2010*; *Khan et al., 2012*; *Stamper et al., 2012*; *Catania, 2013*; *Sponberg et al., 2015*; *Lockey and Willis, 2015*; *Rucci and Victor, 2015*; *Stöckl et al., 2017*). These movements appear unrelated to the movements that are necessary to achieve the task at hand. For example, weakly electric fish will track and stay near a moving refuge, but in addition to the large motions needed to stay near the refuge, there are small whole-body oscillations—an electrosensory analog to microsaccades (*Video 1* and *Figure 1—figure supplement 1*; *Stamper et al., 2012*). Similarly, in behaviors where animals sample discretely over time, animals vary their sampling frequency or the location at which samples are taken, as observed in bats, rats, beaked whales, humans, and pulse electric fish (*Yovel et al., 2010*; *Mitchinson et al., 2007*; *Hartmann, 2001*; *Kothari et al., 2018*; *Caputi et al., 2003*; *Pluta and Kawasaki, 2008*; *Nelson and MacIver, 2006*; *Schnitzler et al., 2003*; *Madsen et al., 2005*; *Yang et al., 2016*; *Hoppe and Rothkopf, 2019*).

There have been several theories proposed to account for these sensing movements including information maximization or infotaxis (*Figure 1*; *Najemnik and Geisler, 2005*; *Vergassola et al.,*

*For correspondence:
maciver@northwestern.edu

Competing interests: The authors declare that no competing interests exist.

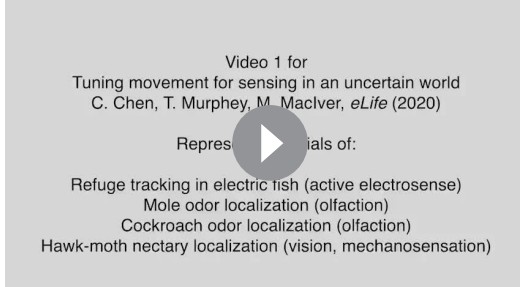

**Video 1.** Segments of behavior across the four species analyzed.

https://elifesciences.org/articles/52371#video1

*2007*; *Yovel et al., 2010*; *Calhoun et al., 2014*; *Álvarez-Salvado et al., 2018*; *Yang et al., 2016*), gain adaptation (*Stöckl et al., 2017*; *Biswas et al., 2018*), spectral whitening (*Rucci and Victor, 2015*), and high-pass filtering (*Stamper et al., 2012*). However, most existing theories are underspecified in that they do not attempt to provide a complete control framework and are therefore incapable of generating realistic trajectories for direct behavioral validation (Discussion). We show that an implementation of energy-constrained proportional betting generates trajectories with good agreement to measured behavior.

Across several methods for computing predictive trajectories of animals tracking targets, an important quantity is the expected information density representing how much information about the state of a target will be gained by moving sensory organs to a given location (EID; *Figure 1A*). Let's assume that we have a way to compute the EID for a given target—a 1-D quantity for a target on a line, a 2-D quantity for a target on a surface as shown in *Figure 1A*, and a 3-D quantity for a target in space. One way to generate target-related behavior is to maximize the information gain over movement, leading to approaching the nearest peak in the EID (*Figure 1B*). A method called infotaxis (*Vergassola et al., 2007*) similarly generates trajectories that maximize expected information by commanding motion toward a peak of the EID. However, expected information maximization leads to problems when there is a high level of uncertainty, as is frequently the case in naturalistic conditions. One problem is that if the gain of information in all directions is low, the prescribed action is to stay in place; yet, animals rarely opt for immobility in the absence of signal. Another problem is susceptibility to distractors. A distractor can be either a real physical object which appears similar to the desired target, or a transient target-like appearance caused by noise. We term these transient appearances fictive distractors to avoid conflation with physical distractors. *Figure 1B* shows the behavior of an expected information maximizing solution in the presence of a distractor. Because the gradient of increased expected information leads to the distractor, the sensor is commanded to go straight to it, potentially ignoring other signal information related to the intended target.

In contrast to the expected information maximizing solution, with energy-constrained proportional betting, sensory organs (or signal emitters in the case of animals like bats and electric fish) are moved to sample spatially distributed signals proportionate to the EID, as shown in *Figure 1C*, balanced by the energetic cost of the movement. The underlying sensory sampling strategy gambles on the chance of obtaining more information at a given location through carefully controlled sensor motion that balances two factors that typically push in opposite directions: (1) proportionally bet on the expected information gain (that is, take more samples by moving slowly in high EID regions and fewer samples by moving more rapidly in low EID regions); and (2) minimize the energy expended for motion. In the example shown in *Figure 1C*, the gamble turns out well since it leads to sampling the location of maximal visibility of the target some distance away from the distractor.

To better communicate our results here, all of which concern localizing a target along a line, we will illustrate energy-constrained proportional betting using a 1-D natural behavior. We use a simplified version of an experiment analyzed later in our study—a hummingbird hawkmoth feeding from a flower swaying laterally in the breeze (*Video 1*, *Figure 1D*; *Sponberg et al., 2015*; *Stöckl et al., 2017*). The moth maintains position by visual and mechanosensory signals. As the moth changes its position with respect to the flower along the flower's line of movement, the signals it uses to localize the flower change. A model relating sensory input values to target locations is called the observation model. A 1-D observation model could be represented by a set of values—first the potential position along the flyway and then the expected sensory signal (of any dimension) given that position. To get the largest signal, a moth should move directly to where it believes the flower is located (peak of the belief in *Figure 1D*). However, to get a better estimate of the flower's location, moving to a position that maximizes the information about the flower location is a better strategy—this position is

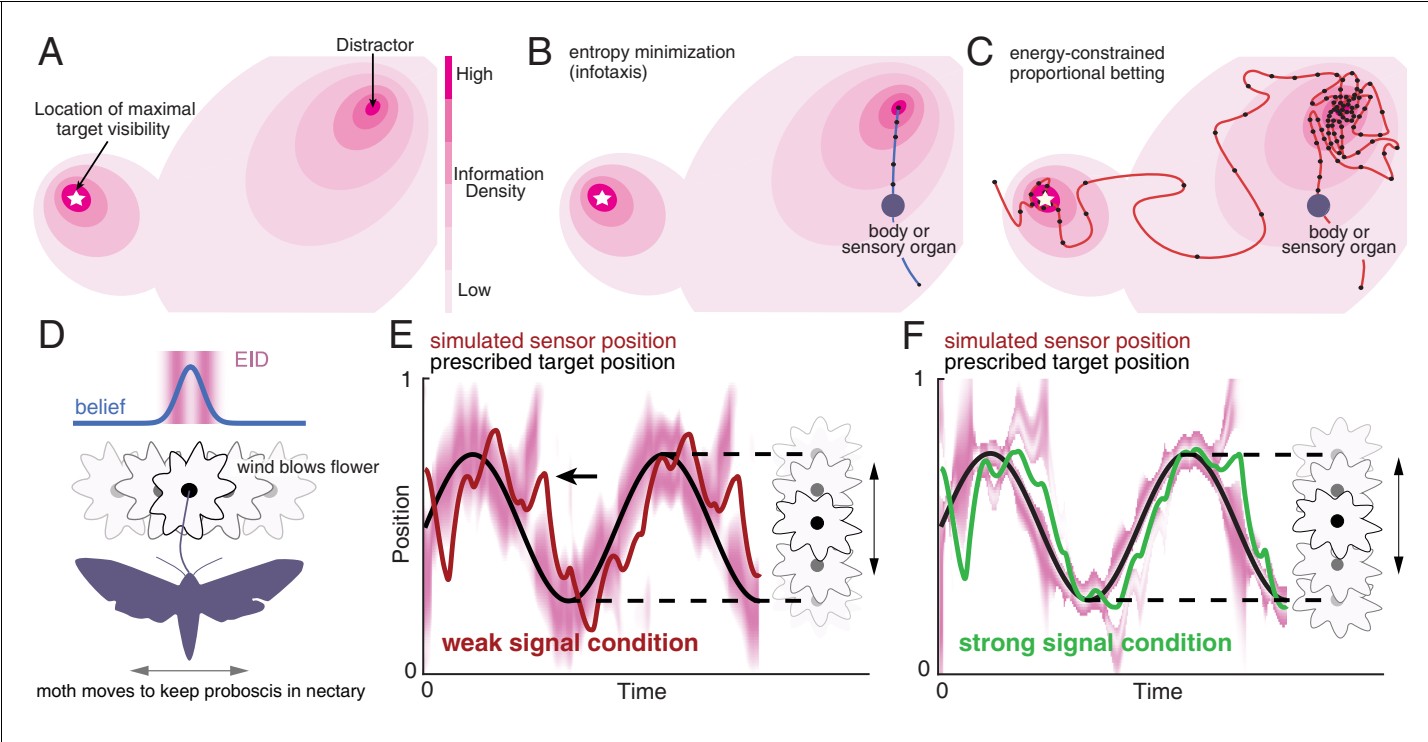

**Figure 1.** Illustration of a 2-D expected information density, information maximization and energy-constrained proportional betting. (A) The heat map represents the expected information density. Because the peak expected information is typically not at the same location as the object, we illustrate the target peak as the point of maximum target visibility. (B) Trajectories generated by information maximization (entropy minimization) locally maximizes the expected information density (EID) at every step, which here commands a path straight to the nearby distractor. In contrast, energy-constrained proportional betting (C) samples the EID proportionate to its density and is balanced by the cost of movement. The natural trade-off between exploration and exploitation that emerges leads to localization of the target and rejection of the distractor (adapted from Figure 3 of *Miller et al., 2016*). Black dots along the trajectory indicate samples at fixed time intervals (longer distances between dots indicate higher speed). (D–F) An illustration of an animal tracking an object constrained to movement in a line, in this case a hypothetical moth using visual signals to track a flower it is feeding from while the flower sways in a breeze in a manner approximated by a 1-D sinusoid—a natural behavior (*Sponberg et al., 2015*). We simulate the tracking of the flower using the ergodic information harvesting algorithm, our implementation of energy-constrained proportional betting. (D) In the top panel, we show an idealization of the moth's belief about the flower's location when the flower reaches the center point (blue line Gaussian distribution above the moth). Higher values in the $y$ direction represents higher confidence of the target at the given $x$ location. The corresponding EID is overlaid in magenta; a darker color indicates higher expected information should the animal take a new sensory measurement in the corresponding location. Note the bimodal structure of this highly idealized EID, with identical maximal information peaks on both sides of the Gaussian belief peak. Intuitively, the expected information is higher along the high slope region of the belief because in this region, small changes in location (x axis direction in the upper inset) is expected to cause large changes in the received sensory signal and hence carry more information about position. (E) Simulation of the moth's position (red curve) while tracking a swaying flower (black curve) in dim light. The corresponding EID is overlaid in magenta. Energy-constrained proportional betting results in persistent activation of movement when the EID is relatively diffuse due to lack of information. Note the presence of a fictive distractor (marked by the black arrow) in the EID due to higher uncertainty in the sensory input as a result of the weaker signal. As seen by the digression at the arrow, ergodic information harvesting (EIH) responded to the distractor by making a detour away from the actual target position to gamble on the chance of acquiring more information but does not get trapped by the distractor because of the proportional betting strategy in combination with the transient nature of fictive distractors. (F) Same as (E), but under strong signal conditions. Now the energy-constrained proportional betting trajectory samples both peaks of the bimodal EID with excursions away from these peaks, similar to measured moth behavior (*Stöckl et al., 2017*, see *Figure 1—figure supplement 1* 'Moth'). Note that even though trajectory segments are planned at 14 fixed-time intervals T (Materials and methods) over the shown duration based on the EID at the start of those intervals, the EID is here plotted continuously for visualization purposes only.

The online version of this article includes the following figure supplement(s) for figure 1:

**Figure supplement 1.** Whole-body or sensory organ small-amplitude motions are ubiquitous as animals track targets.

generally not the same as the estimated flower location (*Yovel et al., 2010*). Instead, places where the change in sensory input is most sensitive to changes in flower location provide the most evidence about the flower's location. The EID will be highest at those places; since those locations are not known, the moth has to compute the EID based on the observation model conditioned on where the flower is expected to be based on current evidence (called its *belief*), giving rise to the two bands of higher expected information density at the maximum slope of the belief (magenta heatmap of *Figure 1D*). In the simulated dim light condition for the moth shown in *Figure 1E*, the EID will be more diffuse due to a more diffuse belief (from higher noise), resulting in larger digressions from the nominal flower trajectory as the moth samples the EID proportional to its density and interrogates distractors (here fictive, arising from random signal fluctuations in a noisy background). In bright light, the EID for the moth is less spread out, and therefore, there is less motion (*Figure 1F*). This illustrates a key behavioral signature of energy-constrained proportional betting—an increase in the magnitude of sensing-related movements as signal weakens or signal noise increases. With no sensory input, there is uniform probability that the target is anywhere in the space, resulting in an energy-constrained trajectory that spans the entire space. This contrasts with the cessation of motion that occurs under very weak signals with an information maximization strategy such as infotaxis (see *Figure 6—figure supplement 1*).

For this study we have quantified the expected information of sensing locations by how much an observation at a location would reduce the Shannon-Weaver entropy (hereafter entropy) of the current estimate of the target's location, as in infotaxis (*Vergassola et al., 2007*). However, other measures of information such as Fisher Information can be used with near identical results (*Miller et al., 2016*). In our approach, the proximity of a given trajectory to perfect proportional betting is quantified by the ergodic metric. The ergodic metric provides a way of comparing a trajectory to a distribution (i.e., in the case of this paper, the EID) by asking whether a trajectory over some time interval has the same spatial statistics as a given distribution (Materials and methods, Appendix 3). Comparing a trajectory to a distribution is a novel capability of the ergodic metric (*Mathew and Mezić, 2011*) that is not shared by common methods of comparing two probability distributions (Appendix 2). Through optimizing a mathematical function that combines ergodicity with the energy of movement (Algorithm 1), we obtain trajectories that bet on information balanced by the metabolic cost to move to informative locations in the space. With a perfectly ergodic trajectory (one with an ergodic measure of zero, only possible with infinite time and when the energy of movement is not constrained), the distribution of expected information is perfectly encoded by the trajectory, or equivalently, the trajectory does perfect proportional betting on the EID. We therefore call the associated algorithm ergodic information harvesting (hereafter EIH, modified from *Miller et al., 2016*, see Materials and methods). *Video 2* provides an animated explanation of EIH in the context of using it to control target localization in an electrosensory robot.

While prior studies have indicated that proportional betting is used at the cognitive decision-making level in primates (*Monosov et al., 2015*; *Gottlieb et al., 2014*), our results suggest that an energy-constrained form of it occurs more broadly as an embodied component of information processing across a wide phylogenetic bracket. Below we will show evidence for this claim by comparing measured tracking trajectories to those simulated with ergodic information harvesting. Our core results use refuge tracking in weakly electric fish, but at the end, we extend our results to three additional previously published datasets encompassing visual and olfactory tracking in insects and mammals.

## Results

First, we present a side-by-side comparison between the one-dimensional tracking trajectories generated by EIH and those we collected from South American gymnotid electric fish (glass knifefish *Eigenmannia virescens*, Valenciennes 1836) as they used electrosense to track a

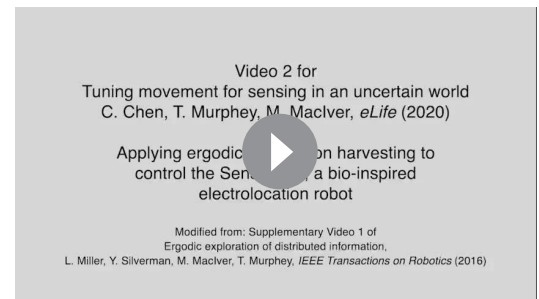

**Video 2.** The ergodic information harvesting algorithm applied to stationary object localization in a bio-inspired electrolocation robot.
https://elifesciences.org/articles/52371#video2

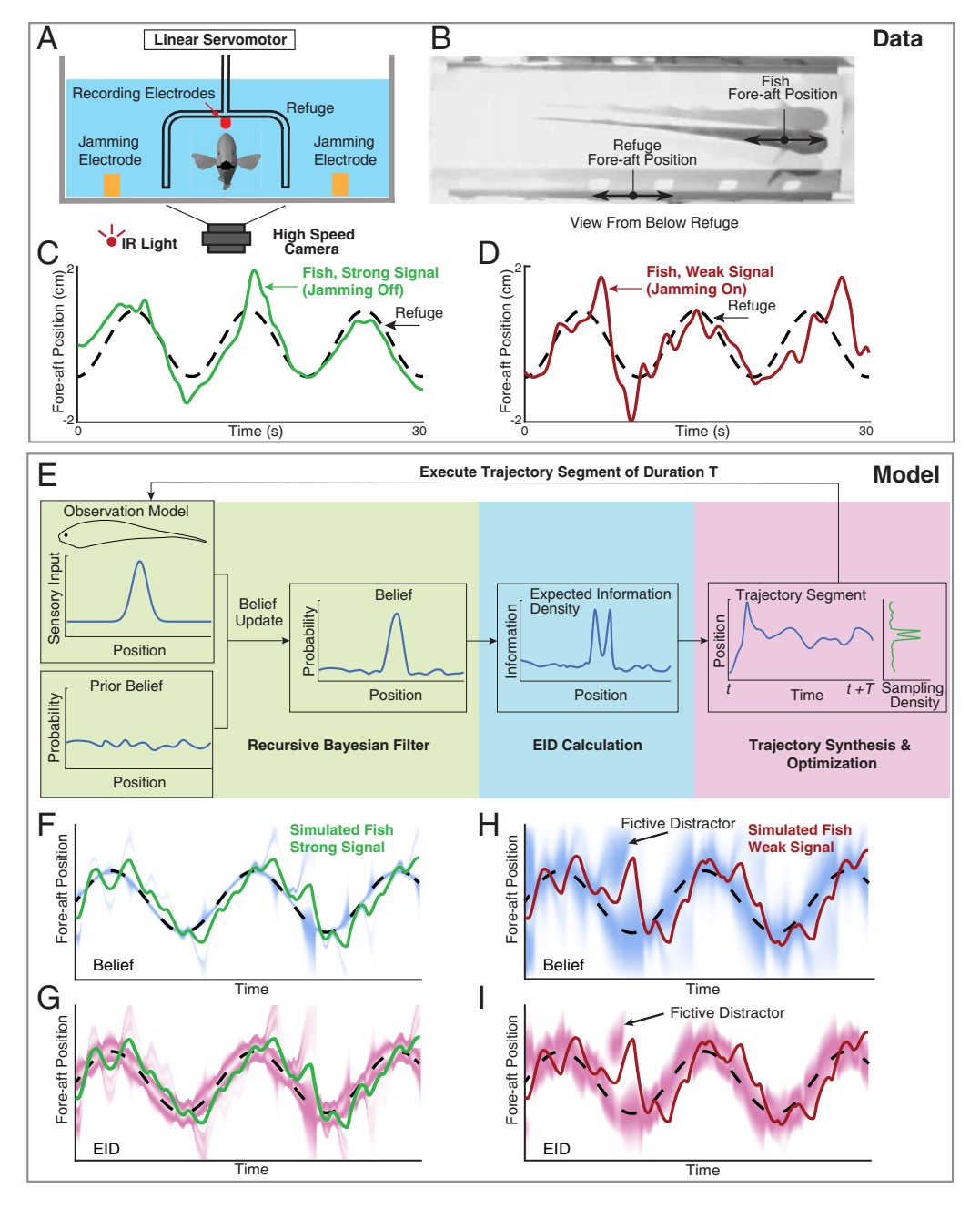

**Figure 2.** Longitudinal refuge tracking behavior in weakly electric fish and core components of EIH. (**A**) Head-on view of experimental apparatus. A computer-controlled linear servo moves the refuge forward and backward along the longitudinal axis of the fish. Jamming electrodes are mounted to the side of the tank and recording electrodes are mounted to the ends of the refuge to generate and monitor the level of electrosensory noise (Materials and methods). (**B**) An example frame of the captured video. (**C–D**) Fish tracking the refuge in the strong signal condition (no jamming applied), and in the weak signal condition (with jamming). Note that departures from the refuge position occur more often and with larger amplitude in the weak signal condition (***Stamper et al., 2012***). (**E**) Core components of the EIH algorithm, shown by colored blocks: a recursive Bayesian filter (green block), an EID calculation (blue block), and synthesis and optimization of a trajectory segment (pink block). The process starts with the simulated fish receiving new sensory input via its position and the observation model. Through the recursive Bayesian filter process, the simulated fish updates its prior belief about the refuge's true location (the belief is initially uniform since the location is unknown) and captures that increase in information in a posterior belief. It then computes the EID based on the posterior belief, explaining why the EID is bimodal when the variance in the estimate is low for the Gaussian observation model: ***Figure 1D and G***. A trajectory segment is then computed that balances proportional betting on the EID (ergodicity) with the cost of movement. The resulting trajectory samples locations proportionate to the EID, as shown by the Sampling Density plot. After these three stages, the simulated fish executes the trajectory segment and returns to begin the process again. (**F, G**) Simulated fish behavior in the strong

*Figure 2 continued on next page*

Figure 2 continued

signal condition. The blue heatmap shows belief (top) of where the target is, where darker colors represent locations with higher probability, and expected information density (bottom) is represented by the magenta heatmap where darker colors represent higher expected information. (H, I) As in F, G, but for the weak signal condition. As in the experimental observations, the departures from the refuge's path are larger and more frequent in the weak signal condition. Note the presence of various color bands representing fictive distractors.

The online version of this article includes the following figure supplement(s) for figure 2:

**Figure supplement 1.** Single target tracking simulation with EIH and infotaxis in the presence of a simulated physical distractor.

**Figure supplement 2.** Dual target tracking simulation with EIH and infotaxis.

**Figure supplement 3.** Effect of jamming and how it varies with jamming intensity.

moving refuge in the dark (*Figure 2A–D*). Second, to examine how well EIH generalizes to other animals with different sensory modalities, we present similar comparisons between EIH and previously published behavioral datasets. These datasets were from blind eastern American moles (*Scalopus aquaticus*, Linnaeus 1758) finding an odor source (*Catania, 2013*); the American cockroach (*Periplaneta americana*, Linnaeus 1758) tracking an odor (*Lockey and Willis, 2015*); and the hummingbird hawkmoth (*Macroglossum stellatarum*, Linnaeus 1758) using vision and mechanosensory cues to track a swaying nectar source while feeding (*Stöckl et al., 2017*). *Video 1* shows excerpts of the behavioral data used for each of the analyses across these species. In all cases, animals were either tracking a moving target (electric fish and moth) or localizing a stationary target (mole and cockroach). Each of the live animal behavior datasets include experiments where the signal versus noise level of the dominant sensory modality driving the behavior was varied. For comparing the resulting trajectories against the predicted trajectory from EIH, this dominant sensory modality was selected for modeling.

Each sensory system was modeled as a 1-D point-sensor with a Gaussian observation model—a deterministic map—that relates the sensory signal value to the variable that the animal is trying to estimate—here assumed to be the position of the target (see Materials and methods). Sensor measurements were simulated by drawing values from the observation model given the sensor's position relative to the target. We added normally distributed measurement noise—also described by a Gaussian function—with variance determined by a specified signal-to-noise ratio (SNR, Materials and methods) to simulate the strong and weak signal conditions present in the live animal trials. These simulated measurements were used to update a probability distribution (often multi-peaked, such as in *Figure 1E* at fictive distractor) representing the simulated animal's belief about the target's likely location through a Bayesian update (*Thrun et al., 2005*) of the previous estimate (Materials and methods). To generate a trajectory for sensory acquisition, at each planning update (Materials and methods), the EIH algorithm takes the updated belief and calculates the expected information density as a function of location (*Figure 2G, I*, the result of the 'EID Calculation' in *Figure 2E*). Then, we generate an ergodic trajectory segment with respect to the EID to simulate the collection of more measurements. Throughout the Results, we show trajectory plots along with the EID heat map—similar to *Figure 2G, I* —to indicate the relationship between sensing-related movements and the EID as EIH carries out proportional betting with respect to the EID.

## Weak signal conditions trigger increased exploratory movement

We first examined the weakly electric fish's tracking behavior under strong and weak signal conditions (*Figure 2A–D*). Weakly electric fish engage in a behavior termed refuge tracking where they try to maintain their position inside a close-fitting open-ended enclosure—such as a plastic tube— even as that enclosure is translated forward and backward along the lengthwise axis of the fish (*Video 1*, *Rose and Canfield, 1993*). Refuge tracking is a natural behavior within protective cover swayed by water flow, such as vegetation or root masses, during the fish's inactive (diurnal) periods in the South American rivers in which they live (*Rose and Canfield, 1993*). Prior work has shown that as sensory input is degraded, these fish will engage in larger full-body excursions from the path taken by the refuge (*Stamper et al., 2012*; *Biswas et al., 2018*; *Rose and Canfield, 1993*). For the trials reported here, all in the dark under infrared illumination, we degraded electrosensory input through varying the intensity of an externally imposed electrical jamming stimulus (Materials and methods) which has previously been shown to impair electrolocation performance (*Watanabe and*

*Takeda, 1963*; *Bastian, 1987*; *Ramcharitar et al., 2005*). Two representative fish behavior tracking trials are shown in *Figure 2C–D*. The weak signal condition (*Figure 2D*) resulted in more body movement during tracking.

In *Figure 2F–I*, we show the corresponding EIH output when EIH is given the same target trajectory under simulated strong and weak signal conditions. In these simulations, although the simulation of the entire fish experiment has the target location provided, the EIH algorithm performing the tracking does not know the target location and is only given simulated sensory observations (see Algorithm 1). The progression of the belief distribution over time is shown in *Figure 2F, G* for strong and weak signal conditions, respectively. Immediately below the belief plot, we show the same trajectory with the corresponding EID visualized as the magenta overlay.

In order to quantify the increase in movement during tracking, we defined a measure termed relative exploration, which is the amount of movement of the body divided by the minimum amount of movement required for perfect tracking (Materials and methods). Under this definition, '1x' relative exploration indicates that the tracking trajectory traveled the same distance as the target trajectory. In the presence of additional exploratory movement, as seen in *Figure 2D*, the relative exploration will exceed '1x'. Across the fish behavior data set, we found a significant increase in relative exploration in the weak signal condition compared to the non-jammed condition (*Figure 3A* upper row, Kruskal-Wallis test, $p<0.001$, $n = 21$). This trend is predicted by EIH, with significantly increased relative exploration as the signal weakens (*Figure 3A* lower row, Kruskal-Wallis test, $p<0.001$, $n = 18$).

## Exploratory movement occurs in a separable frequency band

To further characterize the movement patterns enhanced in the weak signal condition and determine whether the increase in exploration is mainly due to these movements, we performed a spectral

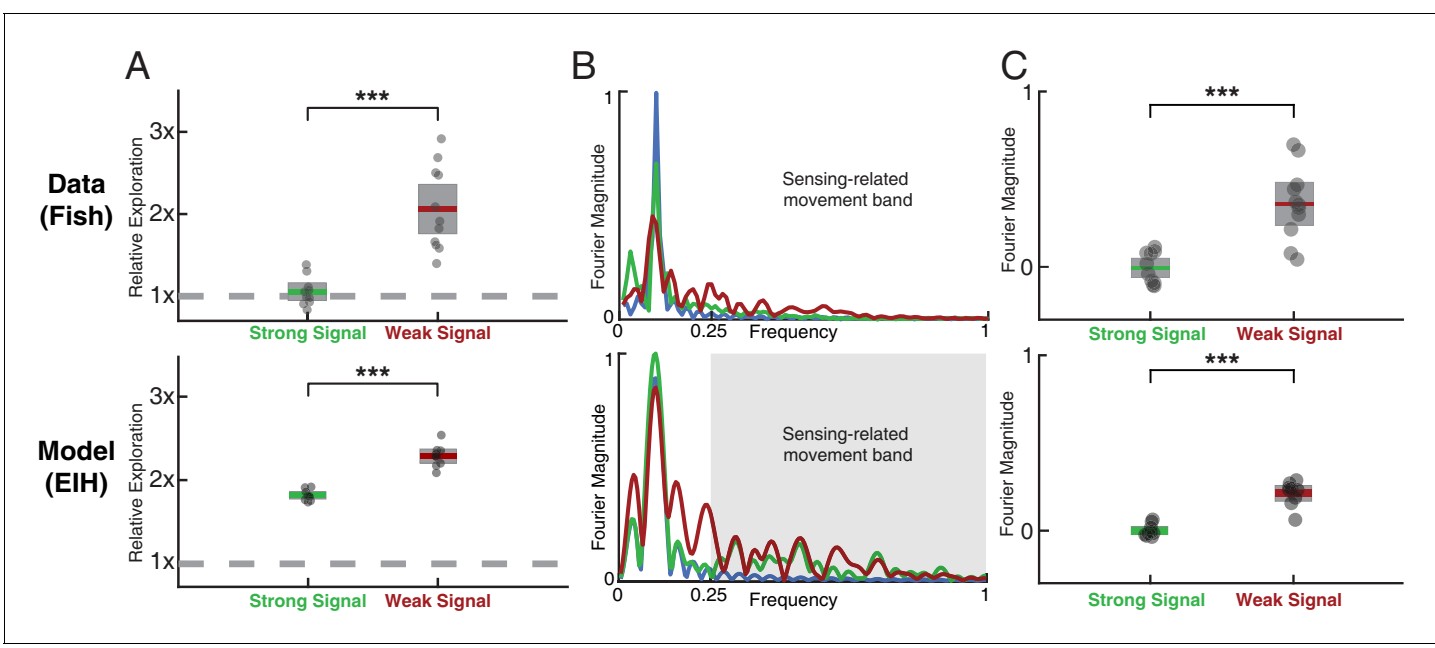

**Figure 3.** Fish behavior versus EIH predictions. (**A**) Relative exploration values (defined in text) for the fish and EIH trajectories under strong and weak signal conditions. Each dot represents a behavioral trial or simulation. EIH (bottom row) shows good agreement with behavioral data (upper row), as both have significantly higher relative exploration in the weak signal condition (Kruskal-Wallis test, $p<0.001$, $n = 21$ for experimental data, and $p<0.001$, $n = 18$ for EIH). (**B**) Representative Fourier spectra of the fish and EIH refuge tracking trajectories as seen in *Figures 2C–D and F–I*, with target trajectory (blue), strong signal condition (green), and weak signal condition (red). The frequencies above the frequency domain of the target's movement are shaded; components in the non-shaded region are excluded in the subsequent analysis. The Fourier magnitude is normalized. (**C**) Distribution of the mean normalized Fourier magnitude within the sensing-related movement frequency band (gray shaded region) for strong and weak signal trials. These distributions are shown after subtraction by the sample mean of the strong signal data to emphasize the difference between strong and weak signal conditions. Each dot represents a behavioral trial or simulation. Significantly higher magnitude is found within the sensing-related movement band under weak signal conditions (Kruskal-Wallis test, $p<0.001$, $n = 21$ for measured behavior, and $p<0.001$, $n = 18$ for EIH). Asterisks indicate the range of $p$ values for the Kruskal-Wallis test (* for $p<0.05$, ** for $p<0.01$, and *** for $p<0.001$).

analysis of the fish's tracking response. In *Figure 3B*, we show the frequency spectrum of the refuge tracking trajectory shown in *Figures 2C–D and F–I*. Two frequency bands can be identified: (1) a baseline tracking band that overlaps with the frequency at which the target (the refuge) was moved; and (2) a frequency band that accounts for most of the increased exploratory movements as signal weakens. We will refer to movements in this separable frequency band as sensing-related movements. The Fourier magnitude is significantly higher for the sensing-related movement frequency band under weak signal conditions when compared to strong signal conditions for both the measured fish behavior (*Figure 3C* upper row, Kruskal-Wallis test, $p<0.001$, $n = 21$) and EIH simulated behavior (*Figure 3C* lower row, Kruskal-Wallis test, $p<0.001$, $n = 18$). This confirms that the significantly increased relative exploration reported in *Figure 3A* is primarily from sensing-related movements rather than baseline tracking movements.

## Sensing-related movements improve refuge tracking performance

A crucial issue to address is whether the additional sensing-related motions measured in the weak signal condition and predicted by EIH cause improved tracking performance. To answer this question, we constructed a filter to selectively attenuate only the higher frequency motion components without affecting the baseline tracking motion (Materials and methods). Simulated weakly electric fish tracking trajectories in the weak signal condition—similar to that shown in *Figure 2H* —were filtered at increasing levels of attenuation. This led to a decrease in sensing-related body oscillations without affecting the baseline tracking motion (pre- and post-filtered trajectories: *Figure 4—figure supplement 1*). Filtered trajectories were then provided as the input to a sinusoidal tracking simulation in which the sensor moved according to the filtered trajectory. With respect to the full EIH sequence shown in *Figure 2E*, the final Trajectory Optimization step was removed, and the trajectory was instead set to the filtered trajectory. The other elements of the EIH algorithm were held constant (Algorithm 1).

We show the results in *Figure 4A* in terms of relative tracking error, where 50% error means a departure from perfect tracking that is one-half the amplitude of the refuge's fore-aft sinusoidal motion. Relative tracking error increases in proportion to the amount of sensing-related motion attenuation, from $\approx 50\%$ with no attenuation to $\approx 75\%$ with the highest attenuation we used. We then evaluated the distance from ergodicity, a dimensionless quantity that measures how well a given trajectory matches the corresponding EID distribution (Materials and methods) for all the trajectories. We found that an increase in attenuation also leads to monotonically increasing distance from ergodicity. This indicates that the filtered trajectories are progressively worse at proportionally betting on information (*Figure 4B*). *Figure 4C* combines these two analyses, demonstrating that the distance from ergodicity is proportional to tracking error.

## Error versus energy expenditure during fish refuge tracking

We estimated the mechanical energy needed to move the fish body along the measured trajectories in comparison to moving the body along the exact trajectory of the refuge and define the ratio between them as the relative energy. Under this definition, any motion beyond baseline tracking will lead to a higher than 1x relative energy. Electric fish are estimated to have needed significantly more mechanical energy during tracking in the weak signal condition compared to the strong signal condition ($\approx 4$x more, *Figure 5A*, Kruskal-Wallis test, $p<0.001$, $n = 21$). We also examined in simulation how tracking error relates to the estimated mechanical energy expended on moving the body, starting with the unfiltered (EIH) trajectories and progressing through higher attenuation levels that gradually eliminate sensing-related movements. This was done by computing the relative energy for the simulation data shown in *Figure 4A*. We found that the tracking error decreased as the relative energy increased, with diminishing returns as the relative energy level neared that needed for the original unfiltered EIH trajectory ($\approx 30$ times the energy needed to move the body along the sinusoidal refuge trajectory, *Figure 5B*).

## EIH predicts measured behavior across other species and sensory modalities

We next evaluated whether EIH predictions can be generalized to the behavior of other animal species using different sensory modalities. To allow comparison to the relative exploration analysis done

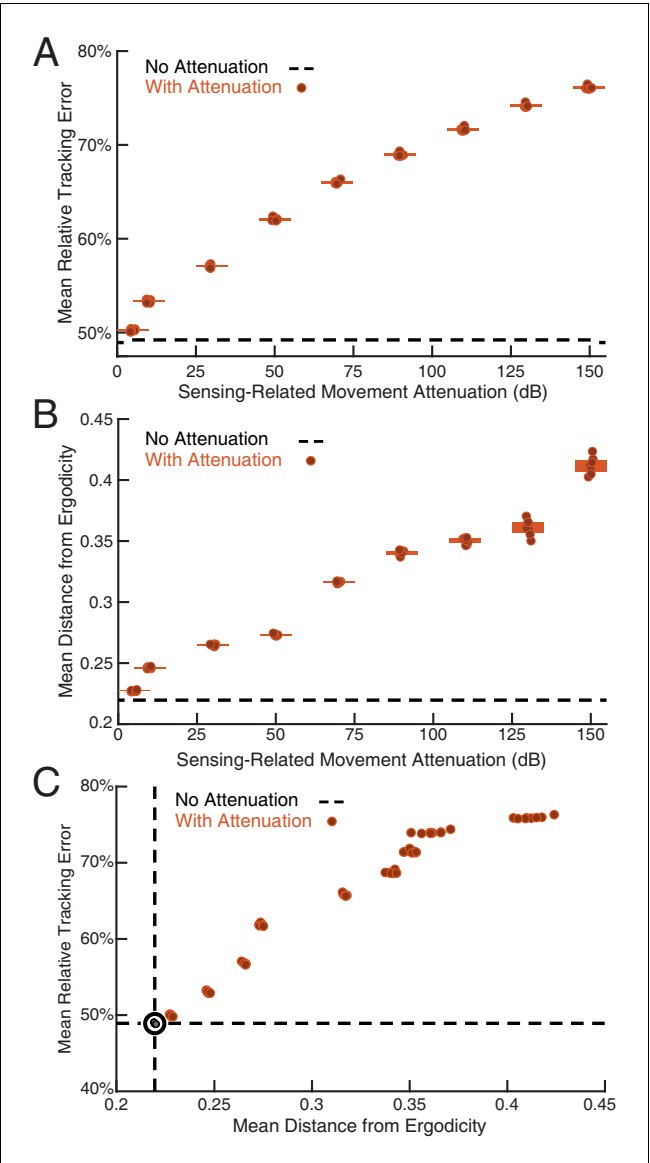

**Figure 4.** Sensing-related movements reduce tracking error. The full-body oscillation in the simulated EIH weak signal sensor trajectory (similar to *Figure 2H*) was gradually removed through stepped increases of attenuation over the sensing-related movement frequency band (Materials and methods) with eight trials per condition (total $n = 80$) to establish the confidence interval. (**A**) Relative tracking error (% of amplitude of the target's fore-aft movement) as sensing-related movement is attenuated. The line near 50% error shows relative tracking error for the original unfiltered trajectory (0 dB attenuation). The thickness of the horizontal bars represents the 95% confidence interval across the eight trials (individual dots) for each attenuation condition. For each attenuation level, the individual trial dots are plotted with a small horizontal offset to enhance clarity. The baseline tracking error with no attenuation is marked by the dashed black line. (**B**) Distance from ergodicity (Materials and methods) as a function of sensing-related movement attenuation. Zero distance from ergodicity indicates the optimal trajectory that perfectly matches the statistics of the EID. As the distance increases from zero, the corresponding trajectory samples the EID further from the perfect proportional-betting ideal. The baseline data with no attenuation is marked by the dashed black line. (**C**) Relative tracking error plotted against distance from ergodicity across attenuation levels. There is a clear positive correlation between tracking error and distance from ergodicity as sensing-related movements are diminished.

The online version of this article includes the following figure supplement(s) for figure 4:

**Figure supplement 1.** How sensing-related movements were attenuated for analyzing the impact of their diminishment.

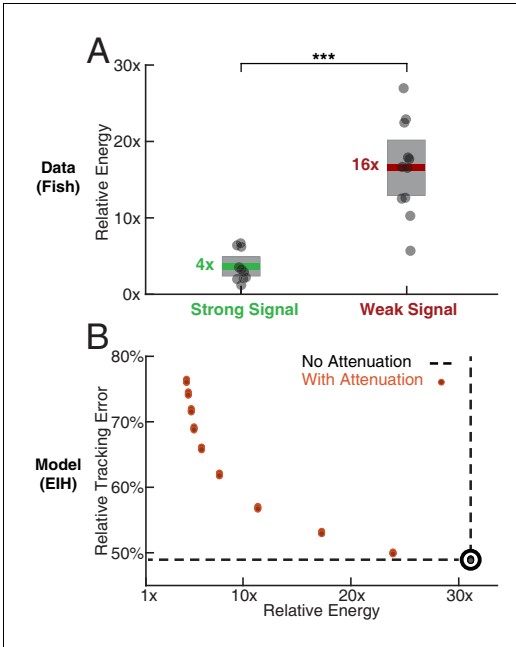

**Figure 5.** Relative energy for electric fish tracking behavior and EIH-generated behavior with attenuated body oscillations. (**A**) Relative energy (definition: text) used by the electric fish during refuge tracking behavior under strong and weak signal conditions. Trials are similar to those shown in *Figure 2C–D*. Weak signal conditions show a significantly higher relative energy as a result of the additional sensing-related movements (Kruskal-Wallis test, $p < 0.001$, $n = 21$). (**B**) Relative energy and relative tracking error for EIH simulations as sensing-related movements are progressively attenuated (data from *Figure 4A–C*, weak signal condition). As the sensing-related movements are attenuated, simulating investment of less mechanical effort for tracking, the relative energy decreases from 30x to less than 5x, but tracking error increases from 50% to around 75%. This plot suggests diminishing returns in tracking error reduction with additional energy expenditure beyond 30x. The lower bound near 4x is similar to the relative energy for the strong signal condition and arises due to small disparities from the sinusoid that the refuge is following (the 1x path). Asterisks indicate the range of $p$ values for the Kruskal-Wallis test (* for $p < 0.05$, ** for $p < 0.01$, and *** for $p < 0.001$).

for the fish, we selected datasets with strong and weak stimulus conditions (see Materials and methods). In *Figure 6—figure supplement 2* we also show an analysis of odor tracking in rats (*Khan et al., 2012*) but excluded from the more thorough analysis performed with the other species considered here due to an insufficient number of trials. *Figure 6* shows each species we considered other than hawkmoth flower tracking (analyzed separately). We include the electric fish for comparison. Below representative tracking trajectories for each species, we show the corresponding EIH-predicted trajectory.

*Figure 6B* shows a representative trial of a mole engaging in a stationary odor source localization task (Materials and methods). The behavioral data show that the mole executes trajectories with significantly larger lateral oscillations under weak signal conditions (normal olfaction degraded by nostril blocking or crossing bilateral airflow, *Catania, 2013*) as summarized in the relative exploration plot (Kruskal-Wallis test, $p < 0.001$, $n = 18$). *Figure 6C* shows a trial of a cockroach localizing an odor source (Materials and methods). Trials under weak signal conditions (normal olfaction degraded by trimming the olfactory antennae length, *Lockey and Willis, 2015*) show an increased amplitude of excursions from the odor track, which leads to a significant increase in relative exploration (Kruskal-Wallis test, $p < 0.002$, $n = 51$).

With respect to relative exploration, EIH shows good agreement with the measured behavior across these species, with significantly increased relative exploration as the signal becomes weak (Kruskal-Wallis test, $p < 0.001$, $n = 18$ for each species). Similarly, good agreement was found for the increase in Fourier magnitude for sensing-related movement frequencies under weak signal conditions compared to strong signal conditions. This is shown in *Figure 7* for the mole data (*Figure 7E–H*, Kruskal-Wallis test, $p < 0.009$, $n = 17$ for measured mole response and $p < 0.001$, $n = 18$ for EIH) and for the cockroach data (*Figure 7I–L*, Kruskal-Wallis test, $p < 0.003$, $n = 51$ for measured cockroach response and $p < 0.001$, $n = 18$ for EIH).

The last species we considered was hawkmoth tracking and feeding from a robotically controlled artificial flower (*Stöckl et al., 2017*). In this case, the investigators used a complex sum-of-sines movement pattern for the artificial flower that is challenging to visualize in the same manner as we have plotted for the target movements used in other species. Instead, we performed a spectral analysis that is similar to the Fourier magnitude analysis. We compared tracking when the moth was under high illumination (strong signal condition) and low illumination (weak signal condition) (*Figure 7M–P*, *Stöckl et al., 2017*). We analyzed the first 18 prime frequency components (up to 13.7 Hz) of both the moth's response (*Figure 7M*, data from *Stöckl et al., 2017*, $n = 13$ for strong signal and $n = 10$ for weak signal) and simulation (*Figure 7N*, $n = 120$ for strong signal and $n = 120$

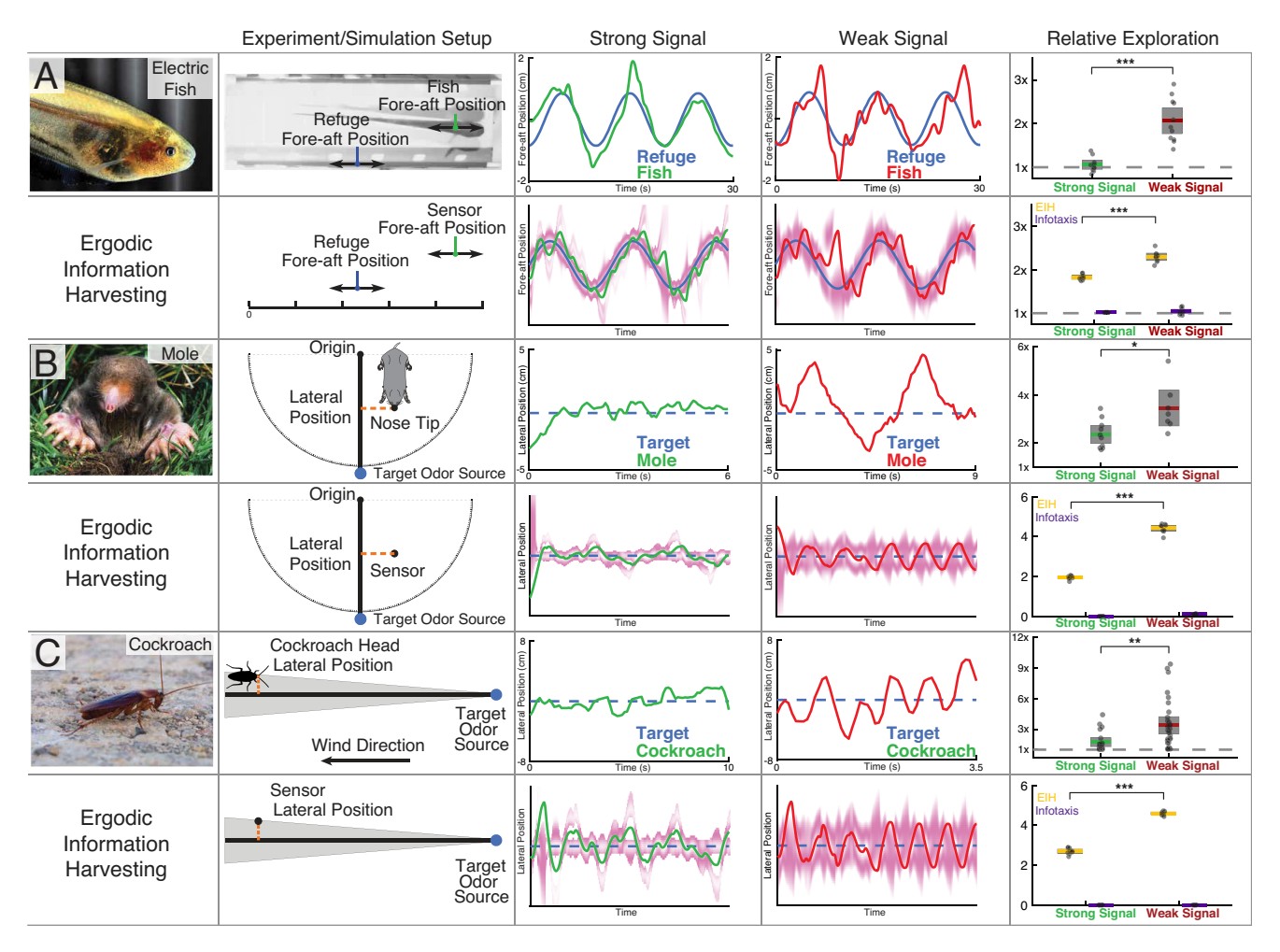

**Figure 6.** Trajectory comparison of animals tracking a target compared to EIH, and relative exploration across all trials. Three representative live animal trajectories above trajectories generated by the EIH algorithm, with their duration cropped for visual clarity. The moth data is not shown here due to the complexity of the prescribed target motion, but is shown in a subsequent figure. All EIH simulations were conducted with the same target path as present in the live animal data, using a signal level corresponding to the weak or strong signal categories (Materials and methods). The EID is in magenta. Relative exploration across weak and strong signal trials (dots) is shown in the right-most column (solid line: mean, fill is 95% confidence intervals). Included for comparison is the relative exploration predicted by infotaxis (*Vergassola et al., 2007*). (**A**) The previously shown electric fish data to aid comparison to other species. As discussed, EIH agrees well with measured tracking behavior. Infotaxis (purple), in contrast, leads to hugging the edge of the EID, resulting in smooth pursuit behavior as indicated by the near 1x relative exploration. (**B**) The experimental setup and data for the mole was extracted from a prior study (*Catania, 2013*). During the mole's approach to a stationary odor source, its lateral position with respect to the reference vector (from the origin to the target) was measured. Relative exploration was significantly higher under the weak signal conditions (Kruskal-Wallis test, $p<0.012$, $n = 17$). In the second row, raw exploration data (lateral distance traveled in normalized simulation workspace units) are shown to allow comparison to simulations, as the latter are done in 1-D (Materials and methods). EIH shows good agreement with significantly increased exploration for weak signal (Kruskal-Wallis test, $p<0.001$, $n = 18$), while infotaxis leads to cessation of movement. (**C**) The experimental setup and data for the cockroach was extracted from a prior study (*Lockey and Willis, 2015*). The cockroach head's lateral position was tracked, and total travel distance was measured during the odor source localization task. Relative exploration is significantly higher for the weak signal condition (Kruskal-Wallis test, $p<0.002$, $n = 51$). In the second row, we show that EIH raw exploration (as defined above) agrees well with measurements as the amount of exploration increased significantly under weak signal conditions (Kruskal-Wallis test, $p<0.001$, $n = 18$), while infotaxis leads to cessation of movement. Asterisks indicate the range of $p$ values for the Kruskal-Wallis test (* for $p<0.05$, ** for $p<0.01$, and *** for $p<0.001$). Photo credit: Copyright Kirk, 2008; Eigenmannia image courtesy of Will Kirk, composite image composed and edited by Eric Fortune and Eatai Roth, under a CC BY 2.5 generic license. © Catania, 2008. Mole image courtesy of Kenneth Catania. Published under a CC BY SA 3.0 unported license. © Wikimedia Commons, 2013. Cockroach image courtesy of Wikimedia Commons. Published under a CC BY SA 3.0 unported license. The online version of this article includes the following figure supplement(s) for figure 6:

**Figure supplement 1.** Systematic comparison between EIH and infotaxis in tracking a sinusoidally moving target.
**Figure supplement 2.** Rat odor tracking behavior and EIH simulation.

*Figure 6 continued on next page*

*Figure 6 continued*

**Figure supplement 3.** Evolution of belief over time for the trials shown in *Figure 6*.
**Figure supplement 4.** Sensitivity analysis on the ratio between control cost and ergodic cost in the objective function of trajectory optimization.
**Figure supplement 5.** Measurements compared to prediction of EIH in two conditions where an animal needs to find the signal during tracking behavior.

for weak signal), which is the same range used in *Stöckl et al., 2017*. We show the spectrum in *Figure 7M–N* as a Bode gain plot rather than Fourier magnitude since the target spectrum covers a wide frequency band including sensing-related movements (Materials and methods). Consistent with previously reported behavior (*Stöckl et al., 2017*), we found significantly increased mean tracking gain in the moth's response within the mid-range frequency region relative to the strong signal condition (*Figure 7O*, Kruskal-Wallis test, $p<0.02$, $n = 23$). This pattern is predicted by EIH simulations with the same sum-of-sine target trajectory (*Figure 7P*, Kruskal-Wallis test, $p<0.001$, $n = 240$).

## Discussion

The body's information processing and mechanical systems have coevolved to afford behaviors that enhance evolutionary fitness. Our theoretical approaches to these domains have proceeded along more independent tracks. Shortly after Shannon published his work on the information capacity of communication channels (*Shannon and Weaver, 1949*), his ideas were applied to visual perception (*Attneave, 1954*; *Barlow, 1959*) to describe efficient coding in the visual periphery. Since then, continual progress has been made in applying information theory to illuminate a host of problems in the coding and energetics of sensory signals from receptors to central nervous system processing (*Atick, 1992*; *Laughlin et al., 1998*; *Niven and Laughlin, 2008*; *Sengupta et al., 2010*). A parallel literature has matured analyzing animal motion (*Waldron et al., 2009*; *Srinivasan and Ruina, 2006*; *Ramdya et al., 2017*; *Nyakatura et al., 2019*; *Aguilar et al., 2016*; *Collins et al., 2005*; *Lee et al., 2008*; *McInroe et al., 2016*; *Sefati et al., 2013*). More recently these two areas are coming together in a growing literature that connects the information gathered through movement to the analysis of movement (*Körding and Wolpert, 2004*; *Cowan and Fortune, 2007*; *Rucci and Victor, 2015*; *Bush et al., 2016*; *MacIver et al., 2010*; *Sprayberry and Daniel, 2007*; *Stamper et al., 2012*; *Biswas et al., 2018*; *Yovel et al., 2010*; *Stöckl et al., 2017*; *Fujioka et al., 2016*; *Yovel et al., 2011*; *Ghose and Moss, 2003*; *Hofmann et al., 2014*; *Bar et al., 2015*; *Nelson and MacIver, 2006*; *Bush et al., 2016*; *Yang et al., 2016*), but a general theory to bridge the gap between the information gained through movement and the energetics of movement is missing (*MacIver et al., 2010*). Energy-constrained proportional betting is a candidate that is sufficiently general to invite application to a host of information-related movements observed in living organisms, while its algorithmic instantiation (Algorithm 1) is able to generate testable quantitative predictions.

In the insects-to-mammals assemblage of animal species analyzed above, we observe gambling on information through motion, where the magnitude of the gamble is indexed by the energy it requires. EIH's approach of extremizing a combination of ergodicity and energy generates trajectories that bet on information, exchanging units of energy for the opportunity to obtain a measurement in a new high-value location. For both measured and EIH-generated trajectories, a key change that occurs as sensory signals weaken is an increase in the rate and amplitude of the excursions from the mean trajectory, which we have quantified as an increase in relative exploration. Although the cause of these excursions in animals is poorly understood, the theory of energy-constrained proportional betting, and the implementation of the EIH algorithm based on that theory, can provide testable hypotheses.

First, the increase in the size of exploratory excursions in weak signal conditions arises with proportional betting because the EID spreads out in these conditions due to high uncertainty (for example, wider magenta bands in *Figure 2I* compared to *Figure 2G*). Because a proportional betting trajectory samples proportionate to the expected information, as the expected information diffuses, the excursions needed for its sampling will correspondingly increase in size.

Second, in EIH, the spectral power profile of these excursions is related to the length of time interval for which a trajectory is generated (variable T in *Figure 2*, see Algorithm 1). One can consider this analogous to how far ahead in time an animal can plan a trajectory before changes in

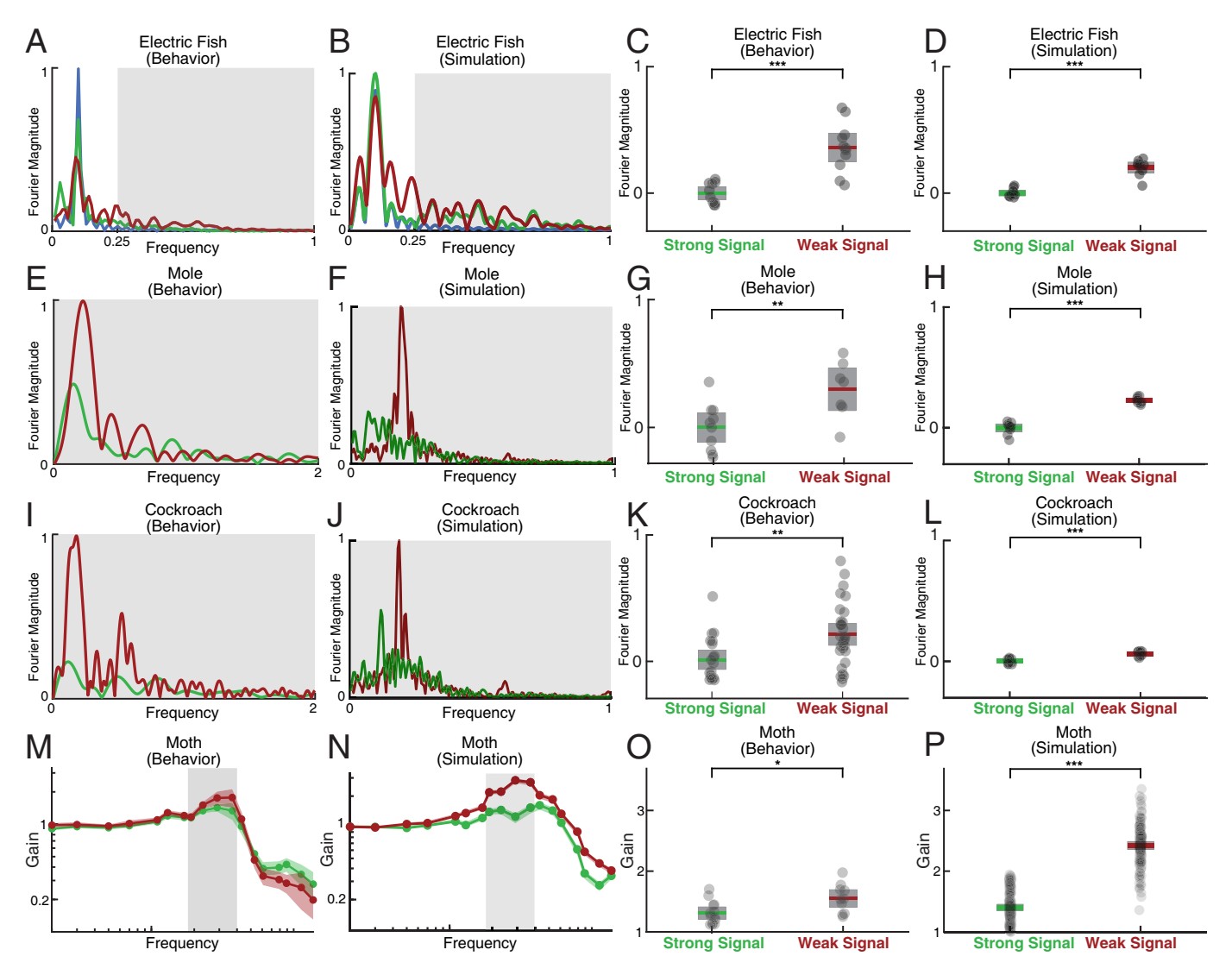

**Figure 7.** Spectral analysis of live animal behavior and simulated behavior. All the single trial Fourier spectra shown in A-B, E-F, and I-J are for the trials shown in *Figure 6*. (A–B) The already shown spectral analysis of the fish tracking data is included here for comparison, with target trajectory (blue), strong signal condition (green), and weak signal condition (red). The frequencies above the frequency domain of the target's movement are shaded; components in the non-shaded region are excluded in the subsequent analysis. The Fourier magnitude is normalized. (C–D) The distribution of the mean normalized Fourier magnitude within the sensing-related movement frequency band (gray shaded region) for strong and weak signal trials, as shown before. (E–F) Representative Fourier spectra of the measured and modeled head movement of a mole while searching for an odor source (lateral motion only, transverse to the line between origin and target), plotted as in A-B. Because the target is stationary in contrast to the fish data, the entire frequency spectrum was analyzed. (G–H) Mean Fourier magnitude distribution, normalized to the strong signal condition to emphasize difference between the strong and weak cases. In the weak signal condition, there is significantly more lateral movement power under weak signal in both the animal data and simulation (Kruskal-Wallis test, $p<0.009$, $n = 17$ for behavior data, and $p<0.001$, $n = 18$ for EIH simulations). (I–J) Fourier spectra of mole's lateral trajectory, plotted the same as in A-B. (K–L) Mean Fourier magnitude distribution. Significantly higher power is found under weak signal in the behavior data and simulations (Kruskal-Wallis test, $p<0.003$, $n = 51$ for behavior data, and $p<0.001$, $n = 18$ for EIH simulations). (M–N) Bode magnitude plot (Materials and methods) of a moth tracking a robotic flower that moves in a sum-of-sine trajectory (figure adapted from *Stöckl et al., 2017*) and the corresponding simulation using the same target trajectory. Each dot in the Bode plot indicates a decomposed frequency sample from the first 18 prime harmonic frequency components of the flower's sum-of-sine trajectory. A total of 23 trials ($n = 13$ for strong signal, and $n = 10$ for weak signal) were used to establish the 95% confidence interval shown in the colored region. Note the increase in gain in the midrange frequency region (shaded in gray) between the strong signal and weak signal conditions. The confidence interval for the simulation Bode plot is established through 240 trials ($n = 120$ for strong signal, and $n = 120$ for weak signal). The same midrange frequency regions (shaded in gray) are used for the subsequent analysis. (O–P) Mean midrange frequency tracking gain distribution for the moth behavior and simulation trials. Moths exhibit a significant

*Figure 7 continued on next page*

*Figure 7 continued*

increase in midrange tracking gain for the weaker signal condition (Kruskal-Wallis test, $p<0.02$, $n = 23$), in good agreement with simulation (Kruskal-Wallis test, $p<0.001$, $n = 240$). Asterisks indicate the range of $p$ values for the Kruskal-Wallis test (* for $p<0.05$, ** for $p<0.01$, and *** for $p<0.001$).

sensory information make planning irrelevant. For example, when tiger beetles see their prey, they execute a trajectory to the prey that is completed regardless of any motion of the prey after initiation of the trajectory. After each segment of running, if they have not caught their prey, they reorient their body toward it and enact a new trajectory, thereby gradually closing the gap (*Gilbert, 1997*). In the strikes of the mottled sculpin, rather than a piece-wise open-loop ballistic strike, the entire strike is ballistic (*Coombs and Conley, 1997*). In contrast, prey strikes in electric fish are ballistic on time scales smaller than 115 ms but adaptive over longer time scales (*MacIver et al., 2001*; *Snyder et al., 2007*). In EIH, over the course of an enacted trajectory segment, changes in the expected information density due to new sensory observations similarly have no effect; these will only be incorporated in the generation of the next trajectory segment.

In EIH, perfect ergodicity is approached through the trade-off between the ergodic metric and the energy of movement within the prescribed trajectory time horizon T. As T asymptotically approaches infinite duration, the system will approach perfect ergodicity as the ergodic measure approaches zero. Conversely, as T asymptotically approaches zero duration, EIH will select a single direction to move to improve information. Changing T between these bounds will affect the frequency components of the sensing-related excursions in the context of exploring a bounded domain while searching for or tracking an object. This is because these excursions originate from the planned trajectory responding to the evolution of the EID, which is sampled at a rate of once every T prior to the synthesis of the next trajectory segment and assumed to be static in between.

For intuition on this point, again consider a fish locating a refuge along a line. As the fish moves to visit a region of high expected information in one direction, the unvisited locations in the other direction start to accumulate uncertainty—the belief distribution will begin to diffuse in those areas at a rate proportional to the noise level. This increase in uncertainty and its relationship to the observation model leads to an increase in expected information in those unvisited locations. This can be seen in *Figure 2I* —the deflection of the trajectory to investigate the fictive distractor present in the EID at the prior trajectory segment results in an increase in the EID in the opposite direction. After the sensor finishes the current trajectory segment of duration T, it then moves in the opposite direction to explore the unvisited regions with high expected information (*Figure 6—figure supplement 3* shows the evolution of belief with respect to the trajectory segment boundaries). Because of these dynamics, a shorter T causes the sensor to react more quickly in response to changes in the EID and hence leads to higher frequency components within sensing-related movements. This same pattern, in combination with EIH's tendency toward sampling across the EID, helps explain why sensing-related movements are often 'oscillatory' (*Stamper et al., 2012*) or 'zigzagging' (*Willis and Avondet, 2005*; *Webb et al., 2004*). The initial T (see *Table 1*) used for the behavior simulations was chosen to fit the frequency of sensing-related oscillations observed in the weakly electric fish refuge tracking data. The same value was applied to mole and cockroach trials and reduced by a factor of five for the moth data due to the higher frequency content of the prescribed robotic flower movement.

As gambling on information through motion involves a trade-off between increasing how well a trajectory approaches ideal sampling (zero distance from ergodicity) and reducing energy expenditure, a useful quantity to examine is how tracking error changes with the energy expended on motion. To do so, we estimated the mechanical energy needed to move the body of the electric fish along the weak and strong signal trajectories, and found that weak signal trajectories required four times as much energy to move the body along as strong signal trajectories. In simulation, we examined how tracking error changes as more energy is invested in sensing-related movements. This analysis shows that the accuracy of tracking increases with the mechanical effort expended on sensing movements, with a 25% reduction of tracking error at the highest level of energy expenditure compared to the low energy case where sensing related movements are removed.

**Table 1.** Parameters of EIH Simulation.

| Parameter | Symbol | Value | Source and note |
|---|---|---|---|
| Variance of observation model | $\sigma_m$ | 0.06 | $\sigma_m$ is initially chosen to fit weakly electric fish behavior and kept the same for all the sensory modalities simulated for the sake of model consistency |
| Time step of the simulation | $\delta_t$ | 0.025, 0.005 | In seconds. $\delta_t$ is initially chosen to fit weakly electric fish behavior and fixed for all the EIH and infotaxis simulations except for moth, where $\delta_t$ is set to 0.005 s to account for the higher velocity of the sum-of-sine trajectory |
| Duration of planned trajectory | T | 2.5, 0.5 | In seconds. T is initially chosen to fit weakly electric fish behavior and kept the same for all the EIH simulations except for moth, where T is set to 0.5 to account for the higher velocity of the sum-of-sine trajectory |
| Step size control of the backtracking line search of trajectory optimization | $\alpha_s$ | 0.1 | $\alpha_s$ and $\beta_s$ are picked to balance between the speed of convergence and the final cost of the trajectory optimization and are fixed across all the EIH simulations |
| Step size control of the backtracking line search of trajectory optimization | $\beta_s$ | 0.4 | $\alpha_s$ and $\beta_s$ are picked to balance between the speed of convergence and the final cost of the trajectory optimization and are fixed across all the EIH simulations |
| Weight of the distance from ergodicity term in the cost function of trajectory optimization loop (see Algorithm 1) | $\lambda$ | 5 | $\lambda$ is initially chosen to fit weakly electric fish behavior and kept the same for all the simulations. Note that changing $\lambda$ changes the trade-off between distance from ergodicity $\mathcal{E}$ (how much information one wants) and control effort (how much energy one is willing to give up). As a result, there is mild sensitivity to this parameter—making it an order of magnitude larger will lead to a more exploratory trajectory while making it an order of magnitude smaller will lead to less exploration. If $\lambda$ is set to zero, no movement will occur at all. For further discussion of this point, see *Miller et al., 2016*. Finally, a sensitivity analysis is also provided in *Figure 6—figure supplement 4* |
| Weight of the control term in the cost function of trajectory optimization loop (see Algorithm 1) | R | 10, 20 | R is initially chosen to fit weakly electric fish behavior and kept the same for all the simulations except for moth, where R is set to 20 since otherwise, the simulated moth body moves faster than the measured data due to the decrease in T from 2.5 to 0.5. Note that the control cost is equivalent to the total kinetic energy required to execute the candidate trajectory given our assumption of a unit point-mass body |
| Number of dimensions used for Sobolev space norm in ergodic metric | $d_S$ | 15 | $d_S$ is initially chosen to be a sufficient number for representing all the behavioral data considered in this paper and kept the same for all the simulations |
| Initial control input | $\alpha(0)$ | 0 | Zero control is applied at the beginning of every simulation |
| Initial belief | $p(\theta_0)$ | $\mathrm{unif}(0,1)$ | Initial belief is set and fixed to a uniform ("flat") prior distribution within the workspace (from 0 to 1) where the probability of the target being at every location is identical |

## Comparison to information maximization

Information maximization and EIH emphasize different factors in target tracking. First, if a scene is so noisy as to have fictive or real distractors, this will generate more than one peak in the probability distribution representing the estimated target location. If the initial location at which information maximization begins is near the wrong peak, information maximization will result in going to that peak (such as to the distractor in *Figure 1B*) and staying at that location. With ergodic harvesting, information across a specified region of interest will be sampled in proportion to its expected magnitude (*Figure 1C*) constrained by the energy expenditure needed to do so. This leads to sensing-related movements that may, at first glance, seem poorly suited to the task: for example, if the location associated with the distractor has higher information density, as it does in *Figure 1C*, then it will be sampled more often than the location associated with the target—but what is important here is that the target is sampled at all, enabling the animal to avoid getting trapped in the local information maxima of the distractor. For information maximization, if 1) there is only one target of interest; 2) the EID is normally distributed; and 3) the signal is strong enough that false positives or other unmodeled uncertainties will not arise, then information maximization will reduce the variance of the estimated location of the single target being sought and direct movement toward the true target location. We interpret the poor agreement between infotactic trajectories and measured behavior as indicating that the conjunction of these three conditions rarely occurs in the behaviors we examined.

Given that these behaviors were all highly constrained for experimental tractability, it seems likely to be even rarer in unconstrained three-dimensional animal behavior in nature.

The second area where these two approaches have different emphases is highlighted in cases where noise is dominating sensory input in high uncertainty scenarios, as is common in naturalistic cases. Information maximization leads to a cessation of movement since no additional information is expected to be gained in moving from the current location (*Figure 6—figure supplement 1A*). Energy-constrained proportional betting will result in a trajectory which covers the space (*Figure 6—figure supplement 1A*): the expected information is flat, and a trajectory matching those statistics is one sweeping over the majority of the workspace at a density constrained by EIH's balancing of ergodicity with energy expenditure. For information maximization, coverage can only be an accidental byproduct of motions driven by information maximization. Appropriate exploration-exploitation trade-offs emerge organically within EIH.

## Other interpretations of the behavioral findings

Fruit bats have been shown to oscillate their clicks from one side to the other of a target, rather than aiming their emissions directly at the target (*Yovel et al., 2010*). *Yovel et al., 2010* suggested that this off-axis sensing behavior arises from the bat aiming a peak in the maximum slope of the signal profile (similar to the behavior of infotaxis, *Figure 1B*). Given that small changes in direction of the sonar beam lead to large changes in echo at the location of maximum slope of the sonar beam distribution, *Yovel et al., 2010* then concludes that the bat is localizing the target optimally. This optimal localization hypothesis is supported by noting that the Fisher information with respect to the sonar beam angle of attack is approximately maximized at the locations where the bats direct the sonar beams. However, this analysis only provides a partial connection to information maximization, because the maximum slope of the sonar beam intensity distribution does not necessarily, or even typically, correspond to a peak in the EID. The main difference between the two is that the EID accounts for the belief distribution—that describes where the target might be and the uncertainty associated with that estimate—whereas the sonar beam distribution only coincides with the EID if one assumes that the ground truth target is located at the peak of the sonar intensity. This means that the bats 'know' where the target is and are using sonar to maintain an estimate rather than globally searching for the target. This interpretation would indicate that the analyzed behavior in *Yovel et al., 2010* is in a late phase of the sonar emission behavior, where the goal is mainly about maintaining an already good estimate of target location. In contrast, EIH performs well in the early stages of search behavior, when the target location is very uncertain and the EID plays a dominant role in behavior. In all the EIH simulations presented here, for example, the simulation began with a uniform prior for the belief, the highest level of uncertainty about target location possible.

Another hypothesis is that active sensing movements arise from the animal adapting its closed-loop tracking gain response to a reduction in signal contrast (*Borst et al., 2005*; *Ghose and Moss, 2006*; *Maimon et al., 2010*; *Biswas et al., 2018*). However, this gain adaptation hypothesis is underspecified, in the sense that critical components are missing to formulate an algorithm that generates predictive trajectories. If gain adaptation is implemented with a Bayesian filter and a process is specified to generate oscillatory motion around targets according to the variance of the belief as a measure of uncertainty, then in the narrow context of a single target with no distractors (neither real nor fictive due to high uncertainty), such an algorithm can be tuned to behave similarly to EIH. However, in more realistic scenarios, there is no apparent mechanism to address real or fictive distractors, a capability of EIH we elaborate on further below. Further work is needed to test the differences between EIH and the gain adaptation hypothesis, or to determine whether gain adaptation is an implementation of EIH in specific, biologically relevant circumstances.

*Khan et al., 2012* show that in rat odor tracking behavior only about 12% of the trajectory qualifies as edge-tracking, suggesting that the rat's zig-zagging trajectory is not centered on the edge of the trail—as predicted by the information maximization hypothesis—but rather on the middle of the odor trail, consistent with ergodic harvesting. They also introduced a model for odor tracking that instructs the sensor to move forward and laterally at a fixed velocity and make decisions to switch the direction of lateral movements based on specific events of sensor measurement and position. Although their model could in principle be adjusted to fit the trajectories of animal tracking under weak signal, their zig-zag sensing-related movements are explicitly programmed to appear based on ad-hoc strategies. This makes the model less generalizable and yields little insight into the

underlying mechanism. In contrast, the sensing-related movements that arise from EIH are not programmed as such but emerge due to the principle of energy-constrained proportional betting. In addition, the Khan et al. model lacks the ability to address distractors, as shown in *Figure 6—figure supplement 3* and *Figure 2—figure supplement 1*, since the movement strategy is not based on the belief or EID map, whereas EIH naturally provides coverage in these scenarios.

Finally, *Rucci and Victor, 2015* and *Stamper et al., 2012* propose that active sensing movements are the outcome of an animal actively matching the spatial-temporal dynamics of upstream neural processing—a process by which the movement serves as a 'whitening filter' (*Rucci and Victor, 2015*) or 'high-pass filter' (*Stamper et al., 2012*). Sensing-related movements could be for preventing perceptual fading (*Kunapareddy and Cowan, 2018*), which has similarities to the high pass filter hypothesis in that motion is to counter sensory adaptation, a high pass filter-like phenomena. Although evidence for the perceptual fading hypothesis during tracking behaviors is lacking, EIH shows good agreement with animal behavior without any mechanism for sensory adaptation. Similar to the gain adaptation hypothesis, the high-pass filter hypothesis is also missing key components for trajectory prediction. Nonetheless, when implemented with the missing components, including a Bayesian filter and a feedback process that generates trajectories that match the desired spatial-temporal dynamics (*Biswas et al., 2018*), the high-pass and whitening-filter hypotheses do not conflict with EIH in single target cases with low uncertainty. This is because EIH also predicts a preferred frequency band for sensing-related movements that may match the preferred spectral power of upstream neural processing. However, in the context of 1) multiple target scenarios; 2) high uncertainty due to weak signal resulting in fictive distractors; or 3) in the absence of any target, the same considerations apply to the high-pass and whitening filter hypotheses as were mentioned for the gain adaptation hypothesis. Further work is needed to test the differences between EIH and the high pass and whitening filter hypotheses, or to determine whether these are an implementation of EIH in specific, biologically relevant circumstances.

## Distractors and multiple targets

Given the above discussion, a capability of EIH that differentiates it from prior theories and that naturally arises from its distributed sampling approach is its ability to reject distractors and sample multiple targets. The live animal experimental data we analyzed did not feature either real distractors (here defined as objects having a distinguishably different observation model from that of the target) or multiple targets (multiple objects with identical observation models). Nonetheless, the EIH simulations suggest that sensing-related motions sometimes occur for rejection of fictive distractors while animals track single objects. A fictive distractor emerges when the current belief for the target's location becomes multi-peaked; each peak away from the true target's location is then a fictive distractor (illustrated by arrow in *Figure 2I*). *Figure 6—figure supplement 3* shows the presence of these fictive distractors in the simulations of the fish, cockroach, and mole tracking behaviors, where we plot the belief rather than the EID. Fictive distractors arise in both the strong and weak signal conditions, but result in small amplitude excursions in the strong signal conditions because of the higher confidence of observations. In simulated tracking behavior, the other source of sensing-related movements beside fictive distractor inspection is the increased spread of the EID as signals weaken, as earlier discussed.

False positive rejection has the signature of a digression from the nominal tracking trajectory; this digression ends when one or more samples have been received indicating there is no object present at the spurious belief peak, which then brings the believed target location back to somewhere closer to the true target position (*Figure 6—figure supplement 3*). In contrast, with a physical distractor, a digression should occur, but the incoming sensory signals support the hypothesis that the object being detected has a different observation model from that of the target, rather than the absence of an object. As none of our datasets include physical distractors, we investigated EIH's behavior in this case with a simulated physical distractor. *Figure 2—figure supplement 1* shows a simulated stationary physical distractor in addition to a stationary target. EIH is able to locate the desired target while rejecting the distractor. This result buttresses a finding in a prior robotics study, where we experimentally tested how EIH responded to the presence of a physical distractor and showed that an electrolocation robot initially sampled the distractor but eventually rejected it (Figure 8 of *Miller et al., 2016*). In comparison, *Figure 2—figure supplement 1* shows that infotaxis stalls as it gets trapped at one of the information maximizing peaks and fails to reject the distractor.

If, instead of a distractor and a target, EIH has two targets, the advantage of EIH's sampling the workspace proportional to the information density is particularly well highlighted. A simulation of this condition is shown in *Figure 2—figure supplement 2*. EIH maintains good tracking with an oscillatory motion providing coverage for both of the targets. As seen in *Figure 2—figure supplement 2*, such coverage is not a feature of infotaxis, which gets stuck at the location of the first target and fails to detect the presence of the other target. A final case to consider is multiple targets with different (rather than identical) observation models. Tracking in such cases requires a simple adjustment to the calculation of the EID that we have explored elsewhere (Equation 13 of *Miller et al., 2016*).

While these preliminary simulations exploring how EIH performs with multiple targets and distractors are promising, it points to a clear need for animal tracking data in the presence of physical distractors or multiple targets (and in 2-D or 3-D behaviors: Appendix 6) in order to better understand whether EIH predicts sensory organ motion better than the gain adaptation or high-pass filtering theories in these cases.

## Biological implementation

The sensor or whole-body sensing-related movements we observe in our results is for proportional betting with regard to sensory system-specific EIDs—for electrosense, olfaction, and vision. To implement EIH, one needs to store at least a belief encoding knowledge about the target. The Bayesian filter update in EIH has the Markovian property, meaning that only the most recent belief is required to be stored. The EID, moreover, is derived from the belief and only used for every generated trajectory segment update, hence does not need to be stored. While the memory needs of EIH are low, trajectory synthesis requires computing the distance from ergodicity between candidate trajectories and the EID, a potentially complex calculation (Materials and methods, Appendix 3). However, the complexity of our calculation may not be indicative of the complexity of implementation in biology. For instance, a recent study (*Stachenfeld et al., 2017*) suggests that a predictive map of future state is encoded in grid cells of the entorhinal cortex through spatial decomposition on a low-dimensionality basis set—a process similar to the calculation of the ergodic metric (Appendix 3). In weakly electric fish, electroreceptor afferents have power law adaptation in their firing rate in response to sensory input (*Drew and Abbott, 2006*; *Clarke et al., 2013*). This makes their response invariant to the speed of the target (*Clarke et al., 2013*) and hence similar to the simulated sensory input used to drive EIH. The power law adaptation also results in a very strong response at the reversal point during whole-body oscillations (Figure 5 of *Clarke et al., 2014*), a response generated by hindbrain-midbrain feedback loops (*Clarke and Maler, 2017*). Given the importance of an increased rate of reversals as signals become weaker in the fish tracking data and EIH simulations, and EIH's invariance to speed, the hindbrain area along with feedback loops to the midbrain are a worthwhile target for future work on the biological basis of EIH.

## Connections to foraging/search theory and stochastic models

Energy-constrained proportional betting trajectories show good agreement with the trajectories of animals whose targets are within sensing range. Foraging theory, such as Charnov's marginal value theorem (*Charnov, 1976*), has modeled animals with random displacements—a situation in which the statistics of resources in the habitat dominates, while whether the resource is within sensing range (or engages learning or memory systems) is not considered to play a role. Lévy walk foraging is only predictive in information-scarce and low resource density contexts (*Wosniack et al., 2017*). This conceptual gap between ecological (animals as stochastic unthinking agents) and neuroscience (animals with sensory systems, memory, and processing) approaches to foraging has a rich historical background (*Hein et al., 2016*; *Mobbs et al., 2018*). Newer work bridges the gap (*Namboodiri et al., 2016*; *Kolling et al., 2012*), such as showing that the turbulent structure of wind-borne odors give rise to Lévy-flight like displacements in seabird navigation (*Reynolds et al., 2015*), that nematode turning movements while looking for food is not stochastic, as previously believed, but rather predictable from recent sensory experience (*Calhoun et al., 2015*), and that planning in volatile environments leverages patchy habitat statistics with processing over the cognitive map and visual sensorium (*Mugan and MacIver, 2020*). Appendix 5 explores connections between our approach and stochastic models more generally, including some limited data indicating

that energy-constrained proportional betting may also predict behavior during searching for targets outside of sensory range (*Figure 6—figure supplement 5*).

## Ergodic movement as an embodied component of information processing

As in the case of fixational eye movements (*Rucci and Victor, 2015*), a common interpretation of body or sensor organ movements away from the assumed singular goal trajectory is that this reflects noise in perceptual or motor processes. Ergodic harvesting presents a competing hypothesis: gathering information in uncertain complex environments means the system should be observed to move away from the singular goal trajectory. These excursions occur as predicted by EIH, including the possibility of multiple targets, and thus increase in size when uncertainty increases.

If an animal is at one peak of a multi-peaked belief distribution, what motivates it to move away from the current peak? The current peak already has sensor noise and other aspects of sensor physics incorporated, but misses other important sources of uncertainty. An occlusion may corrupt signal quality at a location otherwise predicted to have high target information. Other signal generators in the environment may emit confusing signals, or the location may be contaminated by a fictive distractor arising from unmodeled uncertainty. Hence, the opportunity to visit another location in space that is statistically independent—yet contains a similar amount of predicted information—gives an animal an opportunity to mitigate unmodeled uncertainties through the expenditure of energy for movement. This is supported by experiments in human visual search suggesting that saccades are planned in a multi-stage manner for coverage of information towards the task-relevant goal rather than aiming for information maximization (*Yang et al., 2016*; *Hoppe and Rothkopf, 2019*). For example, a model to predict human visual scan paths found 70% of the measured fixation locations were efficient from an information maximization perspective, but there were many fixations ($\approx 30\%$) that were not purely for maximizing information and attributed in part to perceptual or motor noise (*Yang et al., 2016*). We hypothesize that these apparently less efficient fixation locations are in fact the result of gambling on information through motion. It is also possible that motor noise may aid coverage in a computationally inexpensive manner.

The role of motion in this sensing setting is to mitigate the adverse impact of sensor properties. If, however, one is in an uncertain world full of surprises that cannot be anticipated, using energy to more fully measure the world's properties makes sense. This is like hunting for a particular target in a world where the environment has suddenly turned into a funhouse hall of mirrors. Just as finding one's way through a hall of mirrors involves many uses of the body as an information probe—ducking and weaving, and reaching out to touch surfaces—energy-constrained proportional betting predicts amplified energy expenditure in response to large structural uncertainties.

## Materials and methods

### Electric fish electrosensory tracking

Three adult glass knifefish (*Eigenmannia virescens*, Valenciennes 1836, 8–15 cm in body length) were obtained from commercial vendors and housed in aquaria at $\approx 28°C$ with a conductivity of $\approx 100$ μS cm$^{-1}$. All experimental procedures were approved by the Institutional Animal Care and Use Committee of Northwestern University.

An experimental setup was built in which a 1-D robot-controlled platform attached to a refuge allows precise movement of the refuge under external computer control (similar to *Stamper et al., 2012*). The refuge was a customized rectangular section, made by removing the bottom surface of a 15 cm long by 4.5 cm high by 5 cm wide PVC section (3 mm thick) and making a series of 6 openings (0.6 cm in width and spaced 2.0 cm apart) on each side. These windows provide a conductive (water) alternating with non-conductive (PVC) grating to aid electrolocation. The bottom of the refuge was positioned 0.5 cm away from the bottom of the tank to help ensure that the fish stayed within the refuge. A high-speed digital camera (FASTCAM 1024 PCI, Photron, San Diego, USA) with a 50 mm f/1.2 fixed focal length lens (Nikon, Tokyo Japan) was used to capture video from below the tank viewing up at the underside of the fish (*Figure 2B*). Video was recorded at 60 frames s$^{-1}$ with a resolution of 1024 × 256 pixels. The refuge was attached to a linear slide (GL20-S-40-1250Lm, THK Company LTD, Schaumburg, USA), with a 1.25 m ball screw stroke and a pitch of 40 mm per revolution.

The slide is powered by an AC servomotor (SGM-02B312, Yaskawa Electric Corporation, Japan) and servomotor controller (SGD-02BS, Yaskawa Electric Corporation, Japan). The refuge trajectory was controlled by a remote MATLAB xPC target with a customized Simulink model (MathWorks, Natick, USA).

Before each experimental session, individual fish were placed into an experiment tank with water conditions kept identical to that of their housing aquaria and allowed to 2–4 hr to acclimate. The experiment tank was equipped with the refuge control system, high-speed camera and closed-loop jamming system (see below). Trials were done in the dark with infrared LEDs ($\lambda = 850$ nm) used to provide illumination for the camera. Each trial was 80 s long with the jamming signal only applied after the first 10 s and removed for the final 10 s. A total of 21 trials ($n = 10$ for strong signal and $n = 11$ for weak signal) were used for this analysis. During each trial, the servomotor moved the refuge along a predefined 0.1 Hz sinusoidal fore-aft path with an amplitude of 17 mm.

Video of electric fish refuge tracking was processed by a custom machine vision system written in MATLAB to obtain the fish head centroid and location of the refuge at 60 Hz. The $x$ (longitudinal) position of the centroid of head of the fish was filtered by a digital zero-phase low-pass IIR filter with a cut-off frequency of 2.1 Hz and then aligned with the refuge trajectory. For all the completed trials ($n = 21$) across a total of 3 electric fish, the trajectory of both the fish and the refuge trajectory was used for the frequency domain analysis analysis (*Figure 3*). We used the Fourier transform to analyze the fish's tracking response in the frequency domain. Trials with no jamming are categorized as the strong signal condition ($n = 10$, average trial duration 59.6 s). Trials with jamming (jamming amplitude $\geq 10$ mA, see below) are categorized as the weak signal condition ($n = 11$, average trial duration 54.5 s). The cumulative distance traveled by the fish and refuge during refuge tracking was computed and denoted by $D_f$ and $D_r$, respectively. Relative exploration was then defined as $D_f/D_r$.

## Closed-loop jamming system

During refuge tracking, the fish's electric organ discharge (EOD) signal was picked up by two bronze recording electrodes and amplified through an analog signal amplifier (A-M Systems Inc, Carlsborg, USA) with a linear gain of 1000 and a passband frequency from 100 Hz to 1000 Hz. A data acquisition unit (USB 6363, National Instruments, Austin TX, USA) provided digitized signals used in a custom MATLAB script to identify the principal frequency component of the EOD. A sinusoidal jamming signal was then generated through the same USB interface using the digital to analog voltage output channel. The jamming signal's frequency was set to be a constant 5 Hz below the fish's EOD frequency as previously found most effective (*Bastian, 1987*; *Ramcharitar et al., 2005*). The synthesized jamming signal was sent to a stimulus isolator (A-M Systems Inc, Carlsborg WA, USA), which also converted the voltage waveform into a current waveform sent to two carbon electrodes aligned perpendicular to the EOD recording electrodes to avoid interference (*Figure 2A*). The efficacy of the jamming stimulus was verified by examining its effect on the EOD frequency (*Figure 2— figure supplement 3*).

## Mole olfactory localization

Tracking data of blind eastern American moles (*Scalopus aquaticus*, Linnaeus 1758) locating a stationary odor source were digitized from a prior study (*Catania, 2013*). Three experimental conditions were used in the original study: one in which there was normal airflow (categorized as the strong signal condition), one where one nostril was blocked (weak signal condition), and one where the airflow was crossed to the nostril using an external manifold (also weak signal condition). Relative exploration was defined as the ratio between the cumulative 2-D distance traveled by a mole and a straight line from its starting position to the odor location $D_{mole}/D_{line}$ to allow direct comparison between strong and weak signal conditions despite differences in the mole's initial position and target location across trials (*Video 1*, *Figure 6C*). For the corresponding EIH simulation trials, we define relative exploration as the raw cumulative lateral distance traveled by the sensor since the simulation is done in 1-D.

## Cockroach olfactory localization

American cockroach (*Periplaneta americana*, Linnaeus 1758) odor source localization behavior data were acquired from a prior study (*Lockey and Willis, 2015*) through the authors. The cockroach's

head position was tracked during an odor source localization task. The same behavioral experiments were conducted with the odor sensory organ, the antennae, bilaterally cut to a specified length. The control group with intact antennae ($\approx 4$ cm in length) was categorized as the strong signal condition, and the 1 cm and 2 cm antenna-clipped groups were categorized as the weak signal condition (*Video 1*). Only successful trials ($n = 51$, 20 strong signal condition trials and 31 weak signal condition trials) were included in the analysis. Relative exploration for the cockroach data shown in *Figure 6C* is computed as the ratio of the cockroach's total walking distance and the reference path length (the shortest path length from position at the start to the target) $D_{\mathrm{cockroach}}/D_{\mathrm{reference}}$ reported from the study (*Lockey and Willis, 2015*).

## Hawkmoth flower tracking

Hummingbird hawkmoths (*Macroglossum stellatarum*, Linnaeus 1758) naturally track moving flowers in the wind as they insert and maintain their proboscis in the nectary to feed, primarily driven by vision and mechanoreception (*Video 1*, *Sponberg et al., 2015*; *Stöckl et al., 2017*). In a prior study (*Stöckl et al., 2017*), the hawkmoth's flower tracking behavior was measured under various levels of ambient illumination while it fed from an artificial nectary in a robotically controlled synthetic flower. The robotic flower moved in one dimension (lateral to the moth) in a predefined sum-of-sine trajectory composed of 20 prime multiple harmonic frequencies from 0.2 to 20 Hz. The moth's lateral position was tracked, and the Bode gain of the raw tracking data was acquired from a prior study (*Stöckl et al., 2017*). A segment of the moth's raw tracking trajectory is shown in *Figure 1—figure supplement 1*. We classified trials under high illumination (3000 lx, $n = 13$) as the strong signal condition and trials under low illumination (15 lx, $n = 10$) as the weak signal condition.

## Non-technical description of EIH

The animal tracking simulations consist of several components, including simulating the animal moving along a previously synthesized trajectory segment, simulating sensory observations, updating the simulated animal's belief regarding the target's location, computing the EID, and synthesizing the next trajectory segment by optimizing a functional that balances ergodicity with the cost of movement (*Figure 2E*). This algorithmic implementation of energy-constrained proportional betting is built upon a framework we introduced in prior work for stationary target localization by an electrosensory robot using Fisher information (*Miller et al., 2016*). The original algorithm was modified to track moving targets using entropy reduction as the information measure for better comparison to infotaxis, which also used this approach (*Vergassola et al., 2007*) (the results are insensitive to the choice of information metric; near identical results were obtained with Fisher Information). Code to reproduce these simulations, the empirical data, and a Jupyter Notebook tutorial stepping through how the EID is calculated is available (*Chen et al., 2020a*). For pseudocode of EIH and simulation parameters, see Algorithm 1 and *Tables 1–2*.

For each species, we model only one sensory system—the sensory system whose input was degraded through some experimental manipulation during the study. We model the body and sensory system as a unit point-mass sensor moving in one dimension (electrosense for fish, olfaction for mole and cockroach, and vision for moth). The sensory system is assumed to provide an estimate of location, modeled by the observation model which relates raw sensor measurements to the location of the target. Each sensor measurement also includes additive noise modeled by a Gaussian probability distribution with a variance determined by the specified signal-to-noise ratio (SNR) (see next section for more details about how sensory acquisition is simulated). Assuming this observation model and an initially uninformative (uniform) probability distribution of where the target is believed to be (this distribution is called the belief), EIH proceeds as follows: (1) For the current belief, derive the corresponding EID by calculating the answer to the question 'how much information (quantified by entropy reduction) can we obtain by taking a new observation at this location?' for all possible locations (see the 'EID Calculation' block in *Figure 2E*); (2) run the trajectory optimization solver to generate a trajectory segment with duration T (*Table 1*) that optimally balances energy expenditure against the distance from ergodicity with respect to the EID (the focus of this paper and the part of this animal simulation that is specific to this study; see the 'Trajectory Synthesis and Optimization' block in *Figure 2E*); (3) execute the generated trajectory, allowing the sensor to make observations along it (see left half of the 'Recursive Bayesian Filter' block in *Figure 2E*);

**Table 2.** Simulation parameters used for each figure.

| Figure | Category | SNR (dB) | Initial position | Target trajectory | Biological condition |
|---|---|---|---|---|---|
| *Figure 1E* | Weak Signal | ≤30 | 0.7 | Sinusoid | N/A (simulation) |
| *Figure 1F* | Strong Signal | ≥50 | 0.7 | Sinusoid | N/A (simulation) |
| *Figure 2G,I* and *3A-C* | Weak Signal | ≤30 | 0.4 | Sinusoid | N/A (simulation) |
| *Figure 2F,H* and *3A-C* | Strong Signal | ≥50 | 0.4 | Sinusoid | N/A (simulation) |
| *Figure 2—figure supplement 1* | Weak Signal | ≤30 | 0.4 | Stationary | N/A (simulation) |
| *Figure 2—figure supplement 2* | Weak Signal | ≤30 | 0.4 | Stationary | N/A (simulation) |
| *Figure 4A–C* and *Figure 5B* | Weak Signal | ≤30 | 0.4 | Sinusoid | N/A (simulation) |
| *Figure 6A* and *Figure 7B,D* | Strong Signal | ≥50 | 0.4 | Sinusoid | No jamming |
| *Figure 6A* and *Figure 7B,D* | Weak Signal | ≤30 | 0.4 | Sinusoid | ≥ 10 mA jamming |
| *Figure 6B* and *7F,H* | Strong Signal | ≥50 | 0.2 | Stationary | Intact control |
| *Figure 6B* and *7F,H* | Weak Signal | ≤30 | 0.6 | Stationary | Single-side nostril block and crossed airflow |
| *Figure 6C* and *7J,L* | Strong Signal | ≥50 | 0.475 | Stationary | 4 mm intact antenna |
| *Figure 6C* and *7J,L* | Weak Signal | ≤30 | 0.4 | Stationary | 1 and 2 mm bilaterally trimmed antenna |
| *Figure 6—figure supplement 1* | Strong and Weak Signal | 10–55 | 0.4 | Sinusoid | N/A (simulation) |
| *Figure 6—figure supplement 2* | Strong Signal | ≥50 | 0.8 | Prescribed by study (*Khan et al., 2012*) | Sham stitching |
| *Figure 6—figure supplement 2* | Weak Signal | ≤30 | 0.3 | Prescribed by study (*Khan et al., 2012*) | Single-side nostril stitching |
| *Figure 6—figure supplement 3* | Strong Signal | ≥50 | 0.4 | Sinusoid | N/A (simulation) |
| *Figure 6—figure supplement 3* | Weak Signal | ≤30 | 0.4 | Sinusoid | N/A (simulation) |
| *Figure 6—figure supplement 4* | Weak Signal | 20 | 0.4 | Sinusoid | N/A (simulation) |
| *Figure 6—figure supplement 5A* | Weak Signal | ≤30 | 0.9 | Prescribed by study (*Khan et al., 2012*) | Single-side nostril stitching |
| *Figure 6—figure supplement 5B* | Weak Signal | ≤30 | 0.45 | Stationary | Single-side nostril block |
| *Figure 7N,P* | Strong Signal | ≥50 | 0.4 | Prescribed by study (*Stöckl et al., 2017*) | 3000 lux 'high-light' |
| *Figure 7N,P* | Weak Signal | ≤30 | 0.4 | Prescribed by study (*Stöckl et al., 2017*) | 15 lux 'low-light' |

(4) use the incoming observations to update the belief using a recursive Bayesian filter (*Thrun et al., 2005*). This is the step that updates where the simulated animal believes the target is located by taking into account new evidence and existing prior knowledge (see right half of the 'Recursive Bayesian Filter' block in *Figure 2E*); (5) Check whether the termination condition has been reached (either running for a specified time or until the variance of belief is below a certain threshold), and if not, return to step 1. *Video 2* shows these steps graphically for control of a bio-inspired electrolocation robot localizing a stationary target.

## Simulating sensory acquisition for animal tracking simulations

For all analyzed animals, the body or sensory organ being considered is modeled as a unit point-mass in a 1-dimensional workspace. The workspace is normalized to 0 to 1 for all the simulations. Each sensory measurement V is drawn from a Gaussian function that models the signal coming in to

the the sensory system plus a zero-mean Gaussian measurement noise $\epsilon$ to simulate the effect of sensory noise:

$$V = \Upsilon(\theta, x) + \epsilon$$

where $\Upsilon(\theta, x)$ denotes the Gaussian observation model function evaluated at position $x$ given the target stimulus location $\theta$:

$$\Upsilon(\theta, x) = \frac{1}{\sigma_m \sqrt{2\pi}} \exp\left(-\frac{(x-\theta)^2}{2\sigma_m^2}\right)$$

where $\sigma_m$ is the variance of the observation model. Note that the observation model is not a statistical quantity—it is a deterministic map that relates the sensor reading ($\theta \in [0, 1]$) to the location of the target ($x \in [0, 1]$). It should not be confused with the additive noise model which is a statistical quantity described by the Gaussian distribution. If we fix the target location $\theta$ at location 0.5 (center of the normalized workspace) and vary $x$ to take continuous sensory measurements from 0 to 1, the resulting measurement versus location $x$ will form a Gaussian before adding noise (*Figure 1D*, upper inset). The variance $\sigma_n$ of $\epsilon$ is controlled by the signal-to-noise ratio (SNR) of the simulation:

$$\sigma_n = \gamma \sqrt{\frac{\sigma_m^2}{10^{\frac{\text{SNR}_{\text{dB}}}{10}}}}$$

$\sigma_n$ is the variance of the simulated sensory noise $\epsilon$, and $\gamma$ is a unity constant in units of normalized sensor signal unit per normalized workspace unit that converts $\sigma_m$ (in normalized workspace units) on the RHS to normalized sensor signal units of the LHS term $\sigma_n$. We used $\sigma_m = 0.06$ for all simulations (*Table 1*); The SNR values used for all the simulations is documented in *Table 2*. It should be noted that the values of SNR used in EIH simulations are only intended to relate qualitatively ('strong' or 'weak' signal) to the actual (unknown) SNR of the animal's sensory system during behavior experiments. To explore the impact of our assumptions regarding SNR, *Figure 6—figure supplement 1* provides a sensitivity analysis showing how relative exploration varies as a function of SNR.

This generic Gaussian model of observation abstracts the process by which an animal estimates target location from sensory signals. The SNR of the observation model abstracts the effect of endogenous and exogenous noise dilution of sensory signal sources and neurally-derived location estimates in the form of additive zero-mean Gaussian sensory noise. We used 10–30 dB as the weak signal condition and 50–70 dB as the strong signal condition due to the fact that relative exploration plateaus below 30 dB and beyond 50 dB SNR (*Figure 6—figure supplement 1A*). We only intend to use the relative change between high and low SNRs to simulate similar changes in the behavioral conditions of strong and weak signal trials.

The initial condition for all simulations was a uniformly flat (uninformative) belief and an initial state of zero velocity and acceleration. To ensure uniformity, most of the simulation trials were set to have the exact same internal parameters except for SNR, which was changed across trials to compare trajectories in strong and weak signal conditions. The exception was the moth simulations, where we additionally used a smaller value for T, the duration of each planned trajectory segment, and a larger value of R, the weight of the control term in the objective function of the trajectory optimization (a larger R without changing the weight on the energy term $\lambda$ means the trajectories are more energy constrained). A smaller T and larger R was needed due to the significantly higher velocities present in the sum-of-sines trajectory prescribed for the robotic flower in the moth trials (*Table 1*).

## Quantifying information for the expected information density (EID)

Consider the case of an animal tracking a live prey. Suppose that in open space the signal profile of the prey is similar to a 3-D Gaussian centered at its location. For a predator trying to localize the prey, sampling only at the peak of the Gaussian is problematic because while the signal is strongest at that location, it is also locally flat, so small variations in the prey's location has little impact on sensory input. In contrast, any motion of the prey relative to the predator at the maximum slope of the signal profile will result in the largest possible change in the signal received by the predator, and therefore maximize the predator's localization accuracy (*Figure 1D*, the expected amount of information is maximal at the peak of the spatial derivative of the Gaussian profile).

Suppose at time $t$ one has a probability distribution that is the belief $p(x)$ about the value of $x$, for instance about the location of a particular chemical source, prey, or predator. In EIH, a Bayesian filter is used to optimally update $p(x)$ from measurements, so $p(x)$ evolves dynamically over time (**Körding and Wolpert, 2004**). The entropy of $p(x)$, defined by $\sum_i p(x_i) \log(p(x_i))$ (where $i$ is an index over a discretization of the domain), is the amount of information required to describe $p(x)$ as a probability distribution. The entropy of a uniform or flat distribution is high—if it represents object location, it means an object could be at all possible locations in space and requires a lot of information to describe; while a narrow distribution for an object's location can be described with very little information. The EID can be derived by simulating a set of possible sensing locations in the workspace, and for each location predicting the expected information gain by evaluating the reduction in entropy of the posterior with respect to the current prior (Appendix 4).

## Ergodicity

The ergodicity of a trajectory $s(t)$ with respect to a distribution of the information of sensing locations through space $\Phi(x)$ is the property that the spatial statistics of $s(t)$—the regions the trajectory visits and how often it visits them—matches the spatial distribution $\Phi(x)$. Technically, this is quantified by saying that a trajectory $s(t)$ is ergodic with respect to $\Phi(x)$ if the amount of time the trajectory spends in a neighborhood $\mathcal{N}$ is proportional to the measure of that neighborhood $\int_{\mathcal{N}} \Phi(x) dx$ (**Figure 1C**). With a finite time horizon, perfect ergodicity is impossible unless one uses infinite velocity, which motivates a metric on ergodicity (**Scott, 2013**). A metric on ergodicity should be zero when a trajectory is perfectly ergodic and strictly positive and convex otherwise, providing a criterion that can be optimized to make a trajectory as ergodic as possible given the control cost constraint (see below). A standard metric used for comparing distributions is the Sobolev space norm, which can be computed by taking the spatial Fourier transform of $\Phi(x)$ and $s(t)$ (see below). This metric is equivalent to other known metrics such as those based on wavelets (**Scott, 2013**). We can generate an ergodic information harvesting trajectory by optimizing the trajectory with respect to the ergodic metric (**Miller et al., 2016**), often with real-time performance (**Mavrommati et al., 2018**), both in deterministic and stochastic settings (**De La Torre et al., 2016**). See Appendix 1 for background on ergodicity.

## Balancing energy expenditure and proportional betting

In EIH, candidate trajectories are generated (step 2 in the paragraph above) by minimizing the weighted sum of (1) the ergodic metric, which quantifies how well a given trajectory does proportionally betting on the EIH; and (2) the square norm of the control effort. Note that mass is implicitly included in the weighted sum. Optimizing the ergodic metric alone forms an ill-posed implementation as this implies that energy consumption is not bounded. This is equivalent to a situation where the energetic cost of movement is zero, with a consequent movement strategy of sensing everywhere. This is unlikely to be a reasonable movement policy for animals to maximize their chance of survival. Similarly, EIH is not minimizing energy either, as the energy minimizing solution alone is to not move at all. More realistically, when animals have a limited energy budget for movement, the control cost term should be added to impose a bound for energy consumption for a given trajectory. In the first-order approximation of the kinematics of motion of animals, the control cost is defined by the total kinetic energy required to execute the input trajectory (Algorithm 1). In our study, the control cost is not intended to explicitly model the energy consumption of any particular animal used in the study. It is used, however, to represent the fact that energy is a factor that animals need to trade-off with information while generating trajectories for sensory acquisition. The trade-off between ergodicity and the energy of motion is represented by $R/\lambda$, where R is the weight on the control cost and $\lambda$ is the weight on the distance from ergodicity (**Table 1**). For the fish, mole, and cockroach simulations, we used a value of $R/\lambda = 2$, resulting in a relative exploration value of around 2. Due to the higher velocity of the moth's target movement, this was modified to $R/\lambda = 4$ for the corresponding simulations. The variation in relative exploration with an order of magnitude change in $R/\lambda$ from 1 to 10 is 2.5 to 1.5 (**Figure 6—figure supplement 4**).

## Behavioral trajectory simulations

It is worth noting that in EIH, the animal's tracking behavior is hypothesized to be the outcome of a dynamical system, the result of forces and masses interacting, rather than sample paths of a random process—the traditional venue for ergodicity and entropy to play a role in analysis. However, we discuss the possibility that sense organs are moved stochastically in the Discussion and Appendix 5. When used to simulate behavioral trajectories, EIH was reconfigured to use the prescribed stimulus path from the corresponding live animal experiment as the target trajectory (*Figure 6*). The simulated sensor's initial position was set to match the animal's starting location. To simulate the effect of a weak sensory signal, the SNR was reduced in the respective trials to simulate the effect of increased measurement uncertainty. Other than target trajectory, initial position, and SNR, the simulation parameters were the same across all simulations except for moth trials, where T (the duration of each planned trajectory segment) and R (the weight for the control cost term in the trajectory optimization objective function) were adjusted to better fit the higher velocity in the prescribed sum-of-sine target path (*Table 2*).

---

**Algorithm 1. Animal Tracking Simulation with Ergodic Information Harvesting**

---

1: **function** TRIALSIM $(\theta(t), \Upsilon(\theta,x), \sigma_{\mathrm{m}}, \delta_{\mathrm{t}}, T, \alpha_{\mathrm{s}}, \beta_{\mathrm{s}}, \lambda, R, d_{\mathrm{S}}, \alpha(0),$ SNR $)$

•Argument list: ground truth target trajectory $\theta(t)$, observation model $\Upsilon(\theta,x)$, variance of the observation model $\sigma_{\mathrm{m}}$, time step $\delta_{\mathrm{t}}$, length of the planned trajectory segment $T$, step size control of the line search in the trajectory optimization $\alpha_{\mathrm{s}}$ and $\beta_{\mathrm{s}}$, weight for the ergodic metric term in the objective function $\lambda$, weight for the control cost term in the cost function $R$, number of dimensions used for Sobolev space norm in ergodic metric $d_{\mathrm{S}}$, initial control $\alpha(0)$, and SNR of the simulation SNR. See *Table 1* for the value used for each of the parameters, and *Table 2* for the SNR value used for each figure.

•Note: For trial simulations, the target location as a function of time ($\theta(t)$) is set to what it was in the original experimental data set; however, the simulated animal does not know $\theta(t)$. Where EIH is being used in the real world, within a robot or instantiated in biology, $\theta(t)$ would not be specified.

•The observation model is $\Upsilon(\theta,x)$, where $\theta$ is the parameter unknown to the simulated animal (in the tracking context, target location); $x \in \mathcal{X}$ is the position of the sensor ($\mathcal{X}$ is the space of all possible sensor locations in the workspace); measurement $V = \Upsilon(\theta(t),x) + \epsilon$ (each measurement $V$ in simulation is synthesized based on this equation; in real world applications, this is obtained by the animal or robot sensory system), where $V \in \mathcal{V}$ and $\mathcal{V}$ is the set of all possible measurements; and $\epsilon$ is the additive zero-mean Gaussian measurement noise with variance $\sigma_n$ (defined based on the SNR of the simulation and the variance of the observation model $\sigma_{\mathrm{m}}$, see Materials and methods for complete definition).

•Define cost function $J(x,a) = \lambda \mathcal{E}(x(t)) + R \int_0^T \frac{1}{2}a(\tau)^2 d\tau$ for $\lambda > 0$ and $R > 0$, where $\mathcal{E}(x(t))$ is the distance from ergodicity based on the current EID, and $\lambda$ is the ergodic cost weighting factor (see entry for $\lambda$, *Table 1*), while $R \int_0^T \frac{1}{2}a(\tau)^2 d\tau$ is the control cost weighted by $R$ (*Table 1*), $a(\tau)$ is the control input at time $\tau$ that drives sensor motion

•Define initial state , the maximum length of the simulation , and , the duration of each planned trajectory

•Define $\omega > 0$, the threshold on the norm of the gradient used to terminate the line search in the trajectory optimization procedure

•Initialize prior belief as a uniform distribution

2:     Compute initial $\mathrm{EID}(x)$.

3:     **while** $t \leq T_{\mathrm{max}}$ **do**

       Generate an ergodic harvesting trajectory over interval $[t, t+T]$, and then update the belief and EID while executing this trajectory segment

4:         EIH $(p(\theta), \alpha, T, t)$

5:     **end while**

6: **end function**

---

7: **function** EIH $(p(\theta), \alpha, T, t)$

    Argument list: belief $p(\theta)$, control input $\alpha$, length of the planned trajectory $T$, and current time of the simulation $t$.

    Trajectory optimization to solve for the optimal trajectory that minimizes the ergodic measure under the constraint of control cost

8:     **procedure** ERGODIC TRAJECTORY OPTIMIZATION

       Iteratively optimize trajectories (e.g., using gradient descent and $\omega$ to terminate, *Lasdon et al., 1967*) to optimize $J$ at the current time $t$ with the current EID

       **Note**: Singularities in the gradient are handled by adding an arbitrarily small perturbation to the trajectory

9:     **end procedure**

    Execute optimized trajectory, updating the Bayesian belief at every time in the simulation

10:     **procedure** BAYESIAN UPDATE  (see *Thrun et al., 2005*. for more detail on Bayesian filtering)

11:         **for all** $s \in [t, t+T]$, **do**

12:             Simulate system state $(x(s), a(s))$

13:             Take new observation $V(s)$ (in simulation, $V(s)$ is provided to the simulated animal given its $x(s)$ from a synthesis process it does not have access to that uses $\theta(s)$, $\Upsilon(\theta(s),x(s))$, and $\epsilon$; but in real world conditions, $V(s)$ is obtained from the sensory system).

14:             Compute likelihood $p(V \mid \theta)$

15:             Update posterior $p(\theta \mid V)$

16:         **end for**

17:         Compute $\mathrm{EID}(x(t+T))$ for planning the next trajectory segment (see text for details).

           **Note**: For *Figures 1*, *2* and *6*, we plot the EID as it is computed at each time step for illustrative purposes only (thus the EID computation is within the above loop); in either case, the updated EID only impacts the sensor trajectory after $(t + T)$ when the ergodic trajectory optimization routine is called.

18:     **end procedure**

19: **end function**

---

## Sensing-related movement attenuation simulations

The simulated electric fish tracking response under weak signal condition is filtered through zero-phase IIR low-pass filters with different stop band attenuations (*Figure 4—figure supplement 1A*). These filters are configured to pass the low frequency tracking band within ≈0.2 Hz (target motion is a sinusoid in 0.1 Hz). This configuration allows effective removal of higher frequency sensing-related movements without affecting the baseline tracking response. The effect of the sensing-related motion filter is parameterized by the stop band attenuation at 1.5 Hz. The sensing-related movement magnitude can be systematically deteriorated by controlling the stop band attenuation while maintaining intact baseline tracking (*Figure 4—figure supplement 1A–C*).

The raw simulated weak signal tracking trajectory is first filtered by the sensing-related movement filter at stepped attenuation levels from 5 dB to 150 dB. The filtered trajectory is then prescribed to a tracking-only simulation where the sensor is instructed to move along the predefined input trajectory, take continuous sensor measurements, and use these to update the belief and EID. The distance from ergodicity is then evaluated based on the trajectory segment and simulated EID in the same way as for the other behavior simulations. Tracking performance is evaluated by comparing the sensor's best estimate of the target's position over time based on its belief and the ground truth.

## Infotaxis simulations

The original infotaxis algorithm (*Vergassola et al., 2007*) was adapted for 1-D tracking simulations. The infotaxis algorithm computes the EID in the same way as EIH, but differs in the movement policy once the EID is computed. The sensor considers three movement directions from its current position—left, right, or stay—at every planning update. The sensor follows the infotaxis strategy by choosing a movement direction that will maximize the EID and then takes samples along the chosen direction to update the Bayesian filter and consequently the EID for the next planning iteration. The parameters of the infotaxis simulation are kept the same as for EIH to allow direct comparison.

## Energetics

We analyzed how the additional movement for tracking in weak signal conditions affected energy use for electric fish (*Figure 5*). We estimated the net mechanical work required to move the fish along the observed tracking trajectory by first computing the instantaneous power $P(t) = F(t)v(t)$ of tracking at every timestamp. The net force $F(t)$ was estimated by applying Newton's law $F = ma$ with the estimated body mass $m$ (from *Postlethwaite et al., 2009*). Finally, the total mechanical work done by the fish is the integral of the instantaneous power over time $\int_t P(t)dt$. The effect of added mass was included using equations previously developed (*Postlethwaite et al., 2009*).

The relative energy was defined as the total mechanical work of moving the fish along the tracking trajectory divided by the work of moving the fish along the trajectory of the target (the refuge). A relative energy of '1x' therefore indicates that moving the fish along the tracking trajectory required the same energy as moving it along the path that the moving refuge took.

## Spectral analysis

The frequency response of electric fish, mole, cockroach, and moth tracking and simulation data were analyzed using the Fourier transform. The magnitude frequency response data were used in *Figures 3B–C* and *7A–L*. For the 2-D trajectories of mole and cockroach, the lateral tracking response was analyzed separately alongside the 1-D EIH lateral tracking simulation (*Figure 7E–L*). Because our simulations assume a normalized workspace dimension of 0 to 1, the spectral analyses are shown with normalized Fourier magnitudes and are only intended to provide a qualitative link between EIH and animal behavior, rather than matching the units of the original live animal trajectories. For the moth, since the sum-of-sine stimulus covers a wide frequency range that includes the frequencies of the sensing-related movements, the tracking response of moth behavior and simulation is shown in the form of a Bode gain plot (*Figure 7M–N*) instead of Fourier magnitude to visualize both the frequency spectrum of motion and relative exploration of each tracked frequency component. A Bode gain plot shows the magnitude of the frequency response of the tracking

trajectory normalized by the stimulus for a wide range of frequencies. A gain of 1 for any particular frequency indicates the moth (or simulated sensor) responded with the same amplitude as the sum-of-sine stimulus at that frequency. The averaged Fourier magnitude and Bode gain were computed by taking the mean of the Fourier magnitudes or Bode gain within the sensing-related movement frequency window marked by the shaded area of the spectrum plots shown in the first columns of *Figure 7*. For electric fish, the sensing-related movement frequency window is identified as high frequency components outside of the baseline tracking response frequency range (*Figures 3B–C* and *7A–B*). For the mole and cockroach, because the target is stationary and hence there is no baseline tracking frequency, the entire frequency spectrum of the tracking response was used for computing the statistics.

## Quantification and statistical analysis

The Kruskal-Wallis one-way ANOVA test was used for the statistical analysis of relative exploration (*Figures 3* and *6*) and spectral power of tracking (*Figures 3* and *7*). Each trial of weakly electric fish, mole, cockroach, and moth behavior as well as their corresponding simulations were considered independent. Kruskal-Wallis is non-parametric and hence can be applied to test for the significance of relative exploration even though it is a ratio distribution.

The Pearson correlation coefficient and the 95% confidence interval of its distribution were calculated in *Figure 6—figure supplement 1B* based on data from *Figure 6—figure supplement 1A*. The mean and 95% confidence interval was computed for *Figures 3* and *6*, and *Figure 6—figure supplement 1*.

## Data and software availability

The code and data needed to reproduce our results is published separately (*Chen et al., 2020a*). A non-archival online repository is also available (*Chen et al., 2020b*; copy archived at https://github.com/elifesciences-publications/Ergodic-Information-Harvesting). The code includes an interactive Jupyter notebook tutorial on computing the EID. Algorithm 1 provides pseudocode, and *Tables 1– 2* provide the corresponding simulation parameters for the EIH algorithm. *Video 1* shows sample segments of behavioral data for the fish, mole, cockroach and hawkmoth analyses, while *Video 2* provides a graphical explanation of the steps of the EIH algorithm as it is used to control an underwater electrolocation robot.

## Acknowledgements

We thank Mark Willis, Simon Sponberg, and Ken Catania for providing the original behavioral tracking data used for the studies we have cited. We thank the anonymous reviewers for many improvements as well as a suggestion on biological implementation. We thank Madhav Mani and Brennan Sprinkle for helpful discussions and feedback on an earlier draft. Funded by National Science Foundation IIS-1427419, EECCS-1835389, and ECCS-1837515.

## Additional information

### Funding

| Funder | Grant reference number | Author |
| --- | --- | --- |
| National Science Foundation | IIS-1427419 | Malcolm A MacIver |
| National Science Foundation | ECCS-1835389 | Malcolm A MacIver |
| National Science Foundation | ECCS-1837515 | Todd D Murphey |

The funders had no role in study design, data collection and interpretation, or the decision to submit the work for publication.

## Author contributions
Chen Chen, Software, Validation, Methodology; Todd D Murphey, Conceptualization, Supervision, Methodology; Malcolm A MacIver, Conceptualization, Software, Supervision, Funding acquisition, Validation, Visualization, Methodology, Project administration

## Author ORCIDs
Chen Chen (ID) https://orcid.org/0000-0002-3579-0159
Todd D Murphey (ID) https://orcid.org/0000-0003-2262-8176
Malcolm A MacIver (ID) https://orcid.org/0000-0002-3711-8235

## Ethics
Animal experimentation: This study was performed in strict accordance with the recommendations in the Guide for the Care and Use of Laboratory Animals of the National Institutes of Health. Weakly electric fish were handled according to approved institutional animal care and use committee (IACUC) protocol (IS00002740) of Northwestern University.

## Decision letter and Author response
Decision letter https://doi.org/10.7554/eLife.52371.sa1
Author response https://doi.org/10.7554/eLife.52371.sa2

# Additional files
## Supplementary files
• Transparent reporting form

## Data availability
All data and code (v1.0.2) to reproduce these results are archived on Zenodo at http://doi.org/10.5281/zenodo.3988869. An online version of this repository, which may have post-publication corrections, is at https://github.com/MacIver-Lab/Ergodic-Information-Harvesting (copy archived at https://github.com/elifesciences-publications/Ergodic-Information-Harvesting).

The following dataset was generated:

| Author(s) | Year | Dataset title | Dataset URL | Database and Identifier |
|---|---|---|---|---|
| Chen C, Murphey T, MacIver MA | 2020 | Code and data for "Tuning movement for sensing in an uncertain world" (Version v1.0.2) | http://doi.org/10.5281/zenodo.3988869 | Zenodo, 10.5281/zenodo.3988869 |

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

## Appendix 1

### Background on ergodicity

Ergodicity plays an important role in multiple scientific disciplines particularly in stochastic systems and statistical mechanics. In the setting of Markov chains, defined by states and stochastically-driven transitions between states, a system is ergodic if every aperiodic path that leaves a given state must return to that state with probability one (*Lasota and Mackey, 2013*). However, in the present work, we are not interested in stochastic evolutions—though there is the possibility that stochasticity in a system could contribute to coverage needs, something discussed momentarily. Instead, we are interested in deterministic decisions—that is, control decisions—that provide coverage with respect to regions of high information density.

A key insight from *Mathew and Mezić, 2011* was to use the definition of ergodicity for dynamical systems—that a trajectory $x(t)$ spends time in any particular neighborhood $\mathcal{N}$ proportional to the measure of the distribution $\Phi$ over that neighborhood $\int_{\mathcal{N}} \Phi(x)dx$—to create a metric on deterministic trajectories. That is, once $\Phi$ is given, there is nothing stochastic in the question of coverage. Instead, there is only the question of how much coverage a particular trajectory $x(t)$ provides relative to $\Phi$. It should additionally be noted that coming up with a metric is not trivial, in large part because any mathematical comparison must be able to compare two distinct mathematical ideas—a distribution and a trajectory. A distribution is a probability over a region, while a trajectory is a continuum of states parameterized by time $t$. In *Mathew and Mezić, 2011* the authors note two critical steps in creating such a comparison. First, they note that a trajectory can *also* be represented as a sequence of Dirac delta functions $\delta(x)$ also parameterized by time $t$, so that the comparison is between two distributions rather than between a distribution and a trajectory. Secondly, perhaps more importantly, they use the fact that spatial Fourier transforms are well posed for quite general distributions, including Dirac delta functions. From these two steps they conclude that the coefficients of the Fourier transform provide an infinite set of variables that can be used to form a metric. Importantly, none of this analysis requires the trajectories to be stochastic, though similar analysis can be done for stochastic executions (where, for instance, one can imagine that at least small amounts of noise might improve coverage locally).

## Appendix 2

### Relationship between ergodicity and Kullback-Leibler divergence

Ergodicity provides a mathematical approach for comparing a trajectory to a distribution in a way that is similar to how K-L divergence compares a distribution P to a distribution Q. However, K-L divergence cannot be directly applied to trajectories. This is because an idealized trajectory (no uncertainty by itself) is an aggregations of singletons in the form of individual Dirac delta functions (each of zero variance), one for each time $t$. This leads to infinite K-L divergence because the K-L divergence measures how much information changes when using one distribution to represent another distribution (this is why the K-L divergence is often called the relative entropy). In the case of representing a distribution with a delta function, infinite information has been gained because the argument of the delta function is specified with zero variance. As a concrete special case, if one approximates a delta function with normal distributions of decreasing variance, the entropy goes to infinity as the variance goes to zero.

Here is a brief sketch of some of the steps needed to show why K-L divergence will not work for trajectories. This elides a number of technical issues that would need to be carefully worked through for a more rigorous result. Imagine we have a 1-D trajectory that consists of two points (i.e. singletons): $\{0, 1\}$. We can call this a probability density function (PDF) P consisting of two Dirac deltas, $\delta(x - 0)$ and $\delta(x - 1)$—that is, $P(x) = \frac{1}{2}(\delta(x - 0) + \delta(x - 1))$. The differential probabilities $P(0) = P(1)$ both integrate to $\frac{1}{2}$ since the total probability is 1. Suppose we want to compute the K-L divergence between P and Q, where Q is an arbitrary Gaussian distribution with a mean of 0.5 and a non-zero variance. According to the general form of computing K-L divergence:

$$
\begin{aligned}
D_{KL}(P||Q) &= \int_x P(x) \log \frac{P(x)}{Q(x)} dx \\
&= \int_x P(x) \log P(x) dx - \int_x P(x) \log Q(x) dx
\end{aligned}
\tag{1}
$$

where $\int_x P(x) \log P(x) dx$ is the differential entropy of P, which is undefined in this case. To understand why, consider an arbitrary Gaussian distribution, $f(x) \sim \mathcal{N}(\mu, \sigma)$. Computing the first term of the expression for differential entropy (*Equation 1*) gives $h(f) = \int_x f(x) \log f(x) dx = \log(\sigma) + \frac{1}{2} \log(2\pi e)$, which is undefined in the limiting case of a Dirac delta function with $\sigma = 0$ since $\lim_{\sigma \to 0^+} \log(\sigma) = -\infty$.

Hence, the K-L divergence between a Dirac delta function (representing the idealized trajectory) and a smooth (EID) distribution is undefined. (Note that the other term in the K-L divergence that depends on $Q(x)$ will evaluate to a constant, so does not impact the well-posedness of K-L divergence for a trajectory.)

Similar to K-L divergence, mutual information, which quantifies the amount of information obtained about one random variable X by observing another random variable Y, defined as $I(X;Y) = D_{KL}(P(X,Y)||P_X P_Y)$, is another widely used approach for quantifying information between two distributions of random variables. For jointly discrete or jointly continuous pairs $(X, Y)$, it is the K-L divergence between the joint distribution $P(X, Y)$ and the product of the marginal distributions $P_X$ and $P_Y$. Given that the K-L divergence between a trajectory and a distribution is undefined as discussed above, mutual information also cannot be applied to trajectories. More generally, as we stated previously, because the physical trajectories of animals are here considered the behavior of dynamical systems rather than sample paths from a stochastic process, methods like K-L divergence and mutual information that require both inputs to be distributions are undefined and hence will not work in the case where one of these is a trajectory.

## Appendix 3

### How the distance from ergodicity is computed

These details are largely from our prior publication (*Miller et al., 2016*), repeated here for convenience. The spatial statistics of a trajectory $x(t)$ are quantified by the percentage of time spent in each region of the workspace

$$C(x) = \frac{1}{T}\int_0^T \delta[x - x(t)]dt \tag{2}$$

where $\delta$ is the Dirac delta function and T is the duration of the trajectory. The distance from ergodicity is then defined as the sum of weighted squared distances between the Fourier coefficients of the EID $\phi_k$ and the coefficients $c_k$ of the distribution representing the time-averaged trajectory. The Fourier coefficients $\phi_k$ of the distribution $\Phi(x)$ are computed using an inner product

$$\phi_k = \int_X \phi(x)F_k(x)dx \tag{3}$$

and the Fourier coefficients of the basis functions along a trajectory $x(t)$, averaged over time, are calculated as

$$c_k(x(t)) = \frac{1}{T}\int_0^T F_k(x(t))dt \tag{4}$$

where T is the final time and $F_k$ is a Fourier basis function that takes the form of

$$F_k(x) = \frac{1}{h_k}\prod_{i=1}^n \cos\left(\frac{k_i\pi}{L_i}x_i\right) \tag{5}$$

where $h_k$ is a normalization factor (*Mathew and Mezić, 2011*) and $L_i$ is a measure of the length of the dimension. Finally, the ergodic metric is specified as

$$\mathcal{E}(x(t)) = \sum_{k=0}^{\mathbf{K}} \Lambda_k[c_k(x(t)) - \phi_k]^2 \tag{6}$$

where $\mathbf{K}$ is the number of Fourier coefficients used along every one of the $n$ dimensions and $\Lambda_k = (1 + \|k\|^2)^{-s}$, where $\Lambda_k$ is from the Sobolev space norm and $s = \frac{n+1}{2}$ places more weight on lower frequency information (*Mathew and Mezić, 2011*). Given the definition above, the ergodic metric $\mathcal{E}(x(t))$ is the distance from ergodicity, quantifying the difference between a given distribution of EID and the spatial statistics $C(x)$ of a trajectory $x$. We say a trajectory is perfectly ergodic with respect to the EID if $\mathcal{E}(x) = 0$, that is, the spatial statistics of $x$ exactly matches the EID.

## Appendix 4

### How the expected information density (EID) is defined and computed

Given an unknown random variable $\theta$ to estimate (in the context of tracking simulations, $\theta$ represents the location of the tracking target), EIH evaluates an expected information density $\mathrm{EID}(x)$ at every planning update based on the current belief $p(\theta)$. The EID essentially answers the following question: given the probability of $\theta$ being a particular value, and given the likelihood of receiving a particular voltage V corresponding to that value, what is the average amount of information we expect to receive by visiting a state $x$?

Computing $\mathrm{EID}(x)$ requires several steps. First, we define a Gaussian likelihood function that predicts how likely the sensor is to obtain a measurement $V \in \mathcal{V}$ given the current belief $p(\theta)$, where $\mathcal{V}$ is the set of all possible sensor measurements (see Chapter 7.2 of *Robinson, 2016* for details regarding the likelihood function):

$$p(V|\theta,x) = \frac{1}{\sqrt{2\pi}\sigma}\exp\left[-\frac{(V-\Upsilon(\theta,x))^2}{2\sigma^2}\right] \tag{7}$$

Here $x \in \mathcal{X}$ is the location of the sensor ($\mathcal{X}$ is the space of all possible sensor locations), and $\Upsilon(\theta,x)$ is the observation model assuming a known target location $\theta$ evaluated at sensor location $x$.

Next, with a predicted distribution of measurements for each choice of $x$ from the likelihood function $p(V|\theta,x)$, we evaluate what the new posterior belief $p(\theta \mid V,x)$ is expected to be if the sensor were to take a hypothetical measurement at a given location $x$ in the workspace. From the multiplication rule of conditional probability (see Equation 3.16 of *Kokoska and Zwillinger, 2000*),

$$P(A \cap B) = P(A|B)\,P(B) = P(B|A)\,P(A) \tag{8}$$

where A and B are two random events, we obtain the Bayes update rule:

$$P(A|B) = \frac{P(B|A)\,P(A)}{P(B)}. \tag{9}$$

For each choice of potential $x$ where a sensor measurement could be taken, the new posterior is therefore computed by (see Chapter 3.3.9 of *Kokoska and Zwillinger, 2000* and Chapter 2.4 of *Thrun et al., 2005*):

$$\begin{aligned} p(\theta \mid V,x) &= \eta\, p(\theta \mid x)\, p(V \mid \theta,x) \\ &= \eta\, p(\theta)\, p(V \mid \theta,x) \end{aligned} \tag{10}$$

where $\theta$ corresponds to A and V corresponds to B in *Equation 8*. In *Equation 10*, $p(\theta \mid x) = p(\theta)$ because $\theta$ and $x$ are mutually independent, and $\eta = \frac{1}{p(V \mid x)} = \frac{1}{\int_\theta p(V \mid \theta,x)\,p(\theta)d\theta}$ is a normalization factor that constrains the posterior belief $p(\theta \mid V,x)$ to be a probability distribution (see Chapter 2.4 of *Thrun et al., 2005*).

Given a posterior belief $p(\theta \mid V,x)$ evaluated on a potential V measured at a potential location $x$, the entropy reduction from the prior belief $p(\theta)$ can be evaluated using:

$$\Delta S(V,x) = S[p(\theta)] - S[p(\theta \mid V,x)] \tag{11}$$

where $S[p(\theta)] \in \mathbb{R}^1$ and $S[p(\theta)] = -\sum_\theta p(\theta)\log p(\theta)$ is the Shannon-Weaver entropy of the prior belief $p(\theta)$, while $S[p(\theta \mid V,x)]$ is the Shannon-Weaver entropy of the posterior belief.

For any given prior belief $p(\theta)$, the probability of the sensor receiving a measurement V given a choice of sensing location $x$ is not necessarily constant. Therefore, to evaluate the expected entropy reduction at a given sensing location $x$, the entropy reduction $\Delta S(V,x)$ needs to be weighted by the measurement probability $p(V \mid x)$ that is consistent with the prior belief $p(\theta)$. This weighted probability can be obtained by applying the law of total probability (see Equation 3.17 of *Kokoska and Zwillinger, 2000*) to the normalized likelihood function $p(V \mid \theta,x)$ treated as a probability distribution (see Chapter 5.2 of *Robinson, 2016*):

$$p(V \mid x) = \int_{\theta} p(\theta) p(V \mid \theta, x) d\theta \tag{12}$$

Finally, the expected information density at location $x$—$\mathrm{EID}(x)$—is obtained by computing the mathematical expectation (see Chapter 3.5.1 of *Kokoska and Zwillinger, 2000*) of the entropy reduction *if one were* to take a measurement at location $x$. That is, $\mathrm{EID}(x)$ is the weighted average entropy reduction resulting from the conditional probability $p(\theta \mid V, x)$, weighted by the measurement probability $p(V \mid x)$:

$$\mathrm{EID}(x) = \mathrm{E}[\Delta S(x)] = \int_{V} p(V \mid x) \, \Delta S(V, x) dV \tag{13}$$

An interactive Jupyter notebook tutorial on these steps and a video illustrating them graphically is available (*Video 2*, *Chen et al., 2020a*).

## Appendix 5

### Comparison to stochastic models

If animal trajectories are sample paths of a process made up of deterministic and stochastic parts, then some observed small-amplitude oscillations can be modeled by a stochastic search process, similar to that reported in *Drosophila* (*Reynolds and Frye, 2007*; *Censi et al., 2013*; *Mongeau and Frye, 2017*; *Ferris et al., 2018*). Since a sensed target is always present in the tracking behavior analyzed here, it is unlikely that the trajectories analyzed here are purely driven by stochastic search with no intended target (*e.g.* refuge, food source, odor plume). Most models that implement stochasticity by drawing actions based on the EID distribution represent stochasticity in an abstraction of the space in which the body evolves based on its physics (e.g., a Thompson sampling process that randomly samples locations, ignoring the physics and energetics of getting to those locations, *Russo et al., 2017*). If a stochastic signal is directly driving the physics of the body, small random walks will indeed occur, but large-scale motion of the entire body will not occur unless the physical randomness is very large. Moreover, stochastic search can be considered a special case of ergodic search. In general, a random walk will lead to coverage of some area, and that same area could be covered using the ergodic coverage algorithms described here. But the ergodic coverage algorithms enable an animal to adapt as the environment changes, where a given stochastic search will be independent of changes in the environment. This difference may matter in settings where changes in environment matter to search success. Such a scenario is demonstrated in *Figure 6—figure supplement 5* where we provide two examples of target loss. In both cases, the animal exhibits immediate local-to-global search transitions which is naturally reproduced by EIH. Finally, the ergodic harvesting strategy can be applied to stochastic scenarios, where the dynamics include a stochastic process, as shown in *De La Torre et al., 2016*, without substantially impacting the solution.

## Appendix 6

### EIH in 2-D and 3-D behavioral contexts

The presented animal behavior analysis and EIH simulations are limited to a 1-D workspace in the presence of a single target moving along a line. While all animals move in 3-D, the behaviors we examined were minimally distorted by projection to 1-D. Here we consider extensions to behaviors that cannot be projected to 1-D.

Although we show evidence suggesting that EIH naturally balances the exploration-versus-exploitation trade-off in the case of signal loss in 1-D (see *Figure 6—figure supplement 5*), it is unclear how EIH would behave in similar cases in workspaces with more than one dimension. For example, *Calhoun et al., 2014* shows that infotaxis in a 2-D context will respond to a local distractor in the EID by first going straight towards the peak, as also shown in *Figure 1B*, and then engage in circular motion around the distractor peak as it gradually gets rejected by new observations. As a comparison, in *Figure 1C* we show that EIH predicts that the animal will not wander around the distractor peak for long, but rather dwell in such distractor peaks to make additional observations before naturally switching to other regions including the true target location. Such behavior emerges from EIH without any dependence on changes in the EID, whereas infotaxis is dependent on changes in the EID. This also applies to *Figure 6—figure supplement 1* at 20 dB SNR, one of few places where infotaxis exhibits a relative exploration level similar to EIH. This increase in sensor movement is driven by changes in the information landscape due to a very high level of uncertainty. However, the movements of infotaxis at this SNR do not generate systematic coverage with respect to the EID and actually leads to sub-optimal tracking performance ($64 \pm 3\%$ tracking error for EIH versus $74 \pm 4\%$ tracking error for infotaxis at 20 dB; Kruskal-Wallis test, $p<0.001$, $n = 20$). Variations of the EIH algorithm are effective for both higher dimensional systems (*Mavrommati et al., 2018*) and for cooperative systems (*Abraham and Murphey, 2018*). Further investigation is needed to explore the effect of temporal variation in the information landscape on sensing behavior—in 2-D contexts such as explored by *Calhoun et al., 2014* and in 3-D contexts—for insight into how animals approach the exploration-versus-exploitation trade-off in various scenarios such as signal loss.

