## [Decision Letter]

**Acceptance summary:**

This beautifully illustrated manuscript describes a model that captures the trajectories of animal movement during tracking and related exploratory behaviors. The central features of the model are the statistics of information in the environment, and the energetic costs of movement. Simulations using the model produce trajectories that are similar to those produced by animals. This work is a very nice and somewhat provocative contribution to our thinking about active sensing in animals.

**Decision letter after peer review:**

Thank you for submitting your article "Sense organ control in moths to moles is a gamble on information" for consideration by *eLife*. Your article has been reviewed by two peer reviewers, and the evaluation has been overseen by Ronald Calabrese as the Senior and Reviewing Editor. The reviewers have opted to remain anonymous.

The reviewers have discussed the reviews with one another and the Reviewing Editor has drafted this decision to help you prepare a revised submission.

Summary:

The paper by Chen et al. poses a new strategy for interpreting active sensation produced by animal movements. Established in earlier work, the expected information distribution (EID) specifies how future observations would reduce the uncertainty (entropy) of the belief that an agent (animal or robot) has developed about the location of a sensory object. As I understand it, previous work has developed infotaxis, a search algorithm to move towards the position (or other state) where there is a maximum in the EID. Here, the authors identify a problem with this earlier approach that it may allow convergence to distractors and favor locking to an offset point during tracking. The authors use a new strategy (ergodic information harvesting – EIH), posed in an earlier paper, that plans an optimal route through an EID to best reduce uncertainty in the sensory representation while balancing energetic costs. This routing is iteratively refined as the estimate of the object position, the belief, is updated. The authors suggest that this strategy avoids local convergence by encouraging exploration and establishing a tradeoff which may explain the "wiggles" and occasional large divergences animals show

This is a strong presentation and assessment of an interesting hypothesis. The insights that the authors have into active sensing and information acquisition are interesting and represent a clear advance in thinking. The data mining and re-analysis of published literature is clever, carefully done and seems excellent.

Essential revisions:

1) Reviewer #1 provides extensive comments on how the presentation can be revised. "…this manuscript requires significant revision. Rather than focusing on the model and results, most of which have been pushed to the supplemental sections, the manuscript reads as more of a review paper. That the model can be applied to multiple systems is a critical issue, but the manuscript should be more focused on the model rather than the breadth of systems. Further, while I appreciate the scholarship involved in writing the rather broad review of animal movement systems, I found myself constantly asking "what about" other systems and examples that were not covered in the review. This undermines the strength of the argument.

Instead, focus on the model and the remarkable fact that it can be usefully applied to the four systems with appropriate data sets (fish, moles, moths, and roaches). Conclude by saying that this is likely a general mechanism, which can be tested by applying the model to other systems not examined in the current manuscript. It is in this context that you can then usefully review other systems near the end of the manuscript." The comments which follow can guide a revision of the manuscript. We ask the authors to consider this approach. Please provide a rationale for any of the suggested revisions or lack thereof.

2) Reviewer #2 is concerned that Figure 3 be clarified and that "…the authors should make the distinction about the two different EIH conditions (uncertain single targets, and dual targets) more clear in the Introduction." Please address reviewer 2's major comments.

*Reviewer #1:*

The manuscript describes a new model that better captures the trajectories of movements of animals and sensor organs during tracking and exploratory behavior. This model, based on previous work by Miller et al., 2016, posits that 1) animals use movements that are not directly related to achieving the motor task to obtain information and 2) that these movements are controlled in in relation to the statistics of information in the environment and the energetic costs of movement. The main point of the paper is best expressed in a sentence that comes in the last paragraph:

“If, however, one is in an uncertain world full of surprises that cannot be anticipated, using energy to more fully measure the world's properties makes sense.”

The work is, to my knowledge, the first model that generates somewhat realistic trajectories of certain animal movements based on energetics and the quality of sensory information. This is an important contribution for two reasons. A noted in the Discussion, most previous studies of movement related to moment-to-moment control of movement for sensing have been "under-specified" – that is, those models were not designed to simulate movement trajectories. This complete system is a useful addition to the field that I expect will affect how sensing-related movements are studied. Second, as this model generates real trajectories, it can be applied to the control of artificial systems, potentially improving their performance. I am excited by this prospect.

That said, this manuscript requires significant revision. Rather than focussing on the model and results, most of which have been pushed to the supplemental sections, the manuscript reads as more of a review paper. That the model can be applied to multiple systems is a critical issue, but the manuscript should be more focussed on the model rather than the breadth of systems. Further, while I appreciate the scholarship involved in writing the rather broad review of animal movement systems, I found myself constantly asking "what about" other systems and examples that were not covered in the review. This undermines the strength of the argument.

Instead, focus on the model and the remarkable fact that it can be usefully applied to the four systems with appropriate data sets (fish, moles, moths, and roaches). Conclude by saying that this is likely a general mechanism, which can be tested by applying the model to other systems not examined in the current manuscript. It is in this context that you can then usefully review other systems near the end of the manuscript.

My suggestion is that you focus on the analysis of the electric fish data because of the depth of experiments and analysis that was done using these animals in your lab. Please bring some of the supplemental material into the main body of the manuscript. Only after the deep exploration of the fish data you then return to the cross-species analyses, concluding the manuscript with the excellent Figures 3 and 4 of the current manuscript. This will be easier to follow and more convincing than the current organization in which you show the broad application and then focus onto the fish example.

I made some line-by-line comments as I first read the manuscript, listed here.

Abstract: “are actively manipulated throughout stimulus-driven behaviors”

Sensory organs are actively manipulated in non-stimulus driven behaviors as well. Indeed, any movement that an animal makes, regardless of the control regime, manipulates sensory organs due to the simple fact that the receptors are attached to the animal.

I think that what is meant here is that the organs are often moved in ways that are somehow independent of the movements necessary to achieve the task. On one hand, perhaps this is a shorthand way to describe the issue, but it is the first sentence of the manuscript. That said, the subsequent sentences don't make sense if the reader interpreted the first sentence the way I did, so please revise.

Abstract: “While multiple theories for these movements exist…”

In the manuscript I only see a couple of theories – so why not just list them here?

Abstract: “Our approach combines information-theoretic approaches in sensory neuroscience with analyses of the energetics of movement. It can predict sense organ movements in animals and prescribe them in robotic tracking devices.”

This is the real main point of the manuscript – that you've developed a model that captures the tradeoff between information and energetics. Perhaps you can start the manuscript more directly: Movement can be used to obtain information that is not uniformly distributed in the environment. Because movement is energetically costly, there is likely a balance between the benefits of increased sensory information and energetic costs for obtaining that information. We developed a model that can be applied across sensory modalities and species that captures movements for sensing.

Introduction: “small lateral movements”

The movements are not always lateral, even in the examples that were listed here. You can simply omit the word “lateral” or rewrite the sentence.

Introduction: “several models in the literature that have been proposed”

While this is technically a true statement, it is misleading. These models share features but are largely tied to the details of the biological system under study. It might be better to state that there is no generally accepted theory for the control of these sorts of movements. Indeed, the interpretations are linked to the specifics of each model system.

You do not need to reject these ideas to propose your own. Indeed, I think that each of the authors that were cited have already informally considered the idea that there is a limitation for using movement for sensing related to the energetic costs. As stated in the Discussion, most of the previous work is "under-specified" which is to say that those works did not attempt to provide complete control framework as you have done. In this way, it is misleading to suggest some sort of equivalence of those previous models to your own. Your work goes beyond those previous works.

The contribution of this manuscript is a formal description of information gathering via movement that both can be used to analyze biological data and drive the design of artificial systems. Its core proposition is that there is a tradeoff between energy use for movement and the statistics of information generated by that movement.

Introduction paragraph one: “For example, […] Puzzling…”

It is a mistake to say "For example" because this is the start of a directed argument, not just one of the various examples from the literature. What I think that I am reading here is an attempt to "soften the blow" that you deliver here, that you are directly rejecting two hypotheses from the literature – the “signal slope” and “infotaxis” ideas. I don't see these ideas are widely accepted in the literature, and so using them as contrast for your theory is not necessary. I think the manuscript would be stronger if you eliminate this unnecessary background and jump straight to paragraph two: "Here we propose…" The other ideas are reviewed sufficiently in the Discussion.

Figure 1: I did not find this figure to be particularly informative. These traces do not look similar to my eyes, and I think that most people would also struggle to find exactly what compelling information is intended here. The caption states that these trajectories are "strikingly similar" but I can't help but imagine that an equally plausible caption would claim that the trajectories are "strikingly different."

Perhaps you are trying to highlight the fact that there are two categories of movement, task-related which is lower frequency and sensing-related which is higher frequency. But it is difficult to extract that without some stimulus to compare against. And consider, for example, that the cockroach movement which looks like a large oscillation around a mean while the snail has small, high-frequency oscillations with a baseline that is decreasing. And what is the reader expected to see in the nautilus panel? In sum, this figure is "telegraphic" in that the intended interpretation of the figure relies on information that is not available to the reader. Although I suspect that you will want to revise this figure, it is my suggestion that you simply eliminate it. And if you can't bring yourself to eliminate it, please move it to the supplemental.

On a side note – I am worried that the images are unreferenced uses of copyrighted materials from other works. More than worried, in truth. I used one of those images in one of my own efforts and received an e-mail complaint from the person who generated it. That person will certainly read this paper and I suspect be similarly annoyed. Please provide the appropriate citations/references for those images or replace them with your own materials.

Figure 3: Please replace the trajectory example used as the icon for ergotic harvesting. It is confusing because it looks like data. I'm not sure what icon to use for the model instead, but please try something else.

Figure 4: This is a complex figure that, rather than convincing me of the usefulness of the model, instead had me questioning it. In Figure 3, the trajectories are “eyeball-o metrically” convincing. Here in Figure 4 is an approach to quantify that perception. The challenge is that the data sets are quite different from each other, requiring different choices about how to analyze the data. These differences make the analysis appear arbitrary, as each panel is different. Further, the model does not produce output that matches the behavioral data – the data in the right most column do not match the data in the adjacent column. The main finding, that there is an increase in active movements in relation to the strength of the signal, is lost among the many details here.

It is my guess that the result is likely robust to the details of the frequency window, and perhaps you can devise a simpler window or analysis that can be applied to all of the examples, and then show the main result only. Perhaps, for clarity, the Fourier plots could be moved to supplemental for deeper discussion.

Figure S1: I was surprised to find this figure in the supplemental. I think that this should be Figure 1 of the main text as it is the conceptualization of the main contribution of the paper, which is the control model that shows how animals simultaneously manage the uncertain, non-uniform distribution of information in the environment and the costs of movement.

Figure S2: This figure covers an issue that I think will be of interest to most readers, and I also strongly suggest moving this to the main text.

Moving these two supplemental figures to the main text and eliminating the current Figure 1 will focus the manuscript on its important contribution, the model.

OK – to wrap up. I strongly suggest a rewrite with a focus on the model, transforming this paper from reading like a review paper with a model added to a research paper with good scholarship. I would start your edits with the title, which should focus on the main point. I'll hazard a suggestion just to get things rolling – "Tuning movements for sensing in an uncertain world"

“Sense organ control in moths to moles is a gamble on information through motion”

Also, I think that haptic touch and whisking, among the most well-studied systems in which movement is used for sensing, should be better represented in the manuscript.

*Reviewer #2:*

The paper by Chen et al. poses a new strategy for interpreting active sensation produced by animal movements. Established in earlier work, the expected information distribution (EID) specifies how future observations would reduce the uncertainty (entropy) of the belief that an agent (animal or robot) has developed about the location of a sensory object. As I understand it, previous work has developed infotaxis, a search algorithm to move towards the position (or other state) where there is a maximum in the EID. Here, the authors identify a problem with this earlier approach that it may allow convergence to distractors and favor locking to an offset point during tracking. The authors use a new strategy (ergodic information harvesting – EIH), posed in an earlier paper, that plans an optimal route through an EID to best reduce uncertainty in the sensory representation while balancing energetic costs. This routing is iteratively refined as the estimate of the object position, the belief, is updated. The authors suggest that this strategy avoids local convergence by encouraging exploration and establishing a tradeoff which may explain the "wiggles" and occasional large divergences animals show in the trajectories they take.

Overall I find this a very compelling presentation and assessment of an interesting hypothesis. The insights that the authors have into active sensing and information acquisition are interesting and represent a clear advance in thinking. The data mining and reanalysis of published literature is clever, carefully done on the whole and seems excellent. While I have a few comments for clarification, I am enthusiastic about this paper and think it will make a significant contribution to the literature.

The EIH algorithm has been posed earlier, but the authors take a very significant step forward in providing rigorous testing of this hypothesis in the context of single target tracking in several different animals. I especially like the breadth of systems. The prior algorithm development should be more clearly cited in the Introduction.

There is potential concern that the given experiments do not fully reject all alternative models. I don't think this is a deep problem. The authors acknowledge that a gain adjusting infotaxis model might make some of the same predictions at the EIH algorithm and discuss several alternatives. The contrast with fixed gain infotaxis is clear and the alternatives suggest future work to distinguish the "right" modification to simple EID based strategies.

The EIH algorithm seems to make predictions that deviate from infotaxis in two ways. One is when there are multiple targets and one is when there is uncertainty about a target such that a "false" target might be perceived. The authors argue that the latter is what they test and that the former while interesting is beyond the data that they have. I do find the repeated use of the two target EIH simulation a bit distracting when incorporated into the figures with actual animal data (Figure 3) because those experiments don't reflect that part of hypothesis and those specific predictions. This is especially confusing when comparing to tracking data with a moving target. I understand that the second target could be a "false" target rather than a physical multiple target, but it is difficult to gain an intuition that connects the animal data and this example where the second target is a fixed point in space.

In addition to clarifying the figure to avoid confusion, I think the authors should make the distinction about the two different EIH conditions (uncertain single targets, and dual targets) more clear in the Introduction. They can then focus on the former with the animal data, confidently simulate the second, and suggest experiments for the latter as they do in the excellent treatment of this in the Discussion. The way the manuscript is currently framed makes the distractor experiments sound very appealing because they would seem to be strong at distinguishing between the various alternative explanations in the Discussion.

[Editors' note: further revisions were suggested prior to acceptance, as described below.]

Thank you for resubmitting your work entitled "Tuning movement for sensing in an uncertain world" for further consideration by *eLife*. Your revised article has been evaluated by Ronald Calabrese (Senior Editor).

The manuscript has been improved but there are some remaining issues that need to be addressed before acceptance, as outlined below:

The authors have made a strong revision, and they should consider the following feedback from the reviewers to fine-tune the manuscript. The further revision can be made expeditiously and will require only review by the Senior Editor.

*Reviewer #1:*

1) I feel that sections of the Introduction could be shortened or omitted. In the Introduction, most of the paragraph that starts with "An important quantity for implementing energy-constrained proportional betting…" could be deleted. I understand that this example is included to help the reader, but I felt like it didn't add much. It doesn't help because 1) the example of the fish seems clear enough, and 2) adding yet another example, finding a WiFi router, doesn't add much intuition. Indeed, that and the following two paragraphs would, I believe, be clearer if they were shortened and more focussed.

2) In a similar vein, I was surprised by Video 1, which rather than showing data from the manuscript, is a form of tutorial using other (sometimes unrelated) systems and tasks. I think your readers would appreciate videos of the animals in the manuscript – the primary data – perhaps side-by-side with the simulations. As most readers are unlikely to be directly familiar with these behaviors, that would be a contribution. Further, you could assemble video footage for each of the systems that you mention in the paper, something I would like to see! Also, the video is rather uneven in its production, particularly with regards to matching the narration to the video being shown. Please either replace them with data footage, or if you decide to keep the current strategy, spend some more time editing the video content to better match the narration.

Video 2 is a tutorial of the method, which I found (although unusual outside of JoVE) to be useful.

3) Consider adding a paragraph on "Optimal Foraging Theory" in the Discussion. As you know, these models explore the relations between resource distribution, cost of locomotion, and costs of predation (among other factors). These models were particularly popular in the 1990s, involving an approach that is at least parallel to this paper – examining animal behavior in relation to the performance of optimized models. I am not sufficiently familiar with this literature to suggest a paper that generated animal trajectories similar to what was done here. I do recall the Stephens and Krebs book (1986), and I remember reading a paper that had a comparison of locomotor strategies for foraging (https://dx.doi.org/10.1073%2Fpnas.98.3.1089).

4) I didn't mention this in the previous review, but I thought that it might be interesting to add a few sentences in the Discussion about the differences in perspective between your approach and perspective versus our perspective in the Stamper, 2012 active sensing paper. The data on fish tracking are remarkably similar (coincidentally Figure 3B in both your manuscript and our 2012 paper). Our attention was turned towards the brain and controller – proximate mechanisms – whereas the focus here is on “evolutionary” impacts – ultimate mechanisms. This difference in perspective has consequences that may be interesting to discuss. As you know, we did not build a model in the way you did here, but the structure (and implications) of that model would have been quite different. I leave it to you to decide what might be useful (if anything) to add to the Discussion.

*Reviewer #2:*

Overall I think the authors have done an admirable job responding to all of the review comments. I think this paper will make a nice and somewhat provocative addition to the literature. Given the extent of the changes I did have one follow-up comment. The authors have made many changes to the Introduction and I appreciate their efforts. However, I think the change in Figure 1 has gone a little too far in the opposite direction. While I appreciate the change to Figure 1 and the attention to minimizing the discussion of the two target case, it is now quite confusing to have this be the only figure in the Introduction. The Introduction now lacks a figure with strong biological ties especially because the Introduction is framed around an uncertain single target search or tracking problem, but the first figure now is only the two target case. Figure 1—figure supplement 2 is actually discussed more in the Introduction. That figure alone is also not sufficient because of the increased emphasis on the fish work. I really like the two target tracking simulation in Figure 1 and it should be in the paper, but it is confusing as the only motivation figure in the Introduction. Perhaps the easiest solution is to add Figure 1—figure supplement 2 or some of the examples from Figure 1—figure supplement 1 back into main Figure 1 keeping in mind the other reviewer's concern that some of the data in Figure 1—figure supplement 1 (originally Figure 1) were a bit underspecified. Alternatively a more schematic or simulated example of the tracking case could be included and discussed but this seems like more work.

---

## [Author Response]

Reviewer #1:The manuscript describes a new model that better captures the trajectories of movements of animals and sensor organs during tracking and exploratory behavior. This model, based on previous work by Miller et al., 2016, posits that 1) animals use movements that are not directly related to achieving the motor task to obtain information and 2) that these movements are controlled in in relation to the statistics of information in the environment and the energetic costs of movement. The main point of the paper is best expressed in a sentence that comes in the last paragraph:“If, however, one is in an uncertain world full of surprises that cannot be anticipated, using energy to more fully measure the world's properties makes sense.”The work is, to my knowledge, the first model that generates somewhat realistic trajectories of certain animal movements based on energetics and the quality of sensory information. This is an important contribution for two reasons. A noted in the Discussion, most previous studies of movement related to moment-to-moment control of movement for sensing have been "under-specified" – that is, those models were not designed to simulate movement trajectories. This complete system is a useful addition to the field that I expect will affect how sensing-related movements are studied. Second, as this model generates real trajectories, it can be applied to the control of artificial systems, potentially improving their performance. I am excited by this prospect.That said, this manuscript requires significant revision. Rather than focussing on the model and results, most of which have been pushed to the supplemental sections, the manuscript reads as more of a review paper. That the model can be applied to multiple systems is a critical issue, but the manuscript should be more focussed on the model rather than the breadth of systems. Further, while I appreciate the scholarship involved in writing the rather broad review of animal movement systems, I found myself constantly asking "what about" other systems and examples that were not covered in the review. This undermines the strength of the argument.Instead, focus on the model and the remarkable fact that it can be usefully applied to the four systems with appropriate data sets (fish, moles, moths, and roaches). Conclude by saying that this is likely a general mechanism, which can be tested by applying the model to other systems not examined in the current manuscript. It is in this context that you can then usefully review other systems near the end of the manuscript.My suggestion is that you focus on the analysis of the electric fish data because of the depth of experiments and analysis that was done using these animals in your lab. Please bring some of the supplemental material into the main body of the manuscript. Only after the deep exploration of the fish data you then return to the cross-species analyses, concluding the manuscript with the excellent Figures 3 and 4 of the current manuscript. This will be easier to follow and more convincing than the current organization in which you show the broad application and then focus onto the fish example.

We agree with the reviewer’s suggested restructuring and have proceeded with:

1) A complete restructuring of the Results section of the paper with a focus on the weakly electric fish data analysis and EIH’s predictions for the fish data. The cross-species comparison analysis now follows after the fish results and serves as additional analyses to support the claim that EIH’s success in predicting refuge tracking behavior generalizes well to other modalities.

2) Major change to the figures supporting the restructured text, including:

a) Figure 1 is moved to become Figure 1—figure supplement 1;

b) Supplement Figure S1 becomes the new Figure 1 and we have added an additional panel A to illustrate the information landscape so as to make the figure more intuitive;

c) Moved Figure 2 to the Supplement to become Figure 1—figure supplement 2;

d) Added new versions of Figure 2 and 3 that introduce the fish behavior experiments, provides an overview of the steps of EIH, and summarizes the statistics of the weakly electric fish behavior data and EIH model predictions. This supports the focus on the model and its application to weakly electric fish in the Results section;

e) Moved Figure 5 and 6 to be the new Figure 4 and 5 to complete the analysis on weakly electric fish behavior data and EIH predictions;

f) Moved the cross-species analysis Figure 3 and 4 to be the new Figure 6 and 7 following the fish data.

I made some line-by-line comments as I first read the manuscript, listed here.Abstract: “are actively manipulated throughout stimulus-driven behaviors”Sensory organs are actively manipulated in non-stimulus driven behaviors as well. Indeed, any movement that an animal makes, regardless of the control regime, manipulates sensory organs due to the simple fact that the receptors are attached to the animal.I think that what is meant here is that the organs are often moved in ways that are somehow independent of the movements necessary to achieve the task. On one hand, perhaps this is a shorthand way to describe the issue, but it is the first sentence of the manuscript. That said, the subsequent sentences don't make sense if the reader interpreted the first sentence the way I did, so please revise.

We agree with the reviewer and changed the sentence to: “While animals track or search for targets, sensory organs make small unexplained movements on top of the primary task-related motions.”

Abstract: “While multiple theories for these movements exist…”In the manuscript I only see a couple of theories – so why not just list them here?

We agree with the reviewer and have expanded the text to list the alternative hypothesis.

Abstract: “Our approach combines information-theoretic approaches in sensory neuroscience with analyses of the energetics of movement. It can predict sense organ movements in animals and prescribe them in robotic tracking devices.”This is the real main point of the manuscript – that you've developed a model that captures the tradeoff between information and energetics. Perhaps you can start the manuscript more directly: Movement can be used to obtain information that is not uniformly distributed in the environment. Because movement is energetically costly, there is likely a balance between the benefits of increased sensory information and energetic costs for obtaining that information. We developed a model that can be applied across sensory modalities and species that captures movements for sensing.

We agree with the reviewer and have therefore integrated a version of the suggested text into the Introduction.

Introduction: “small lateral movements”The movements are not always lateral, even in the examples that were listed here. You can simply omit the word “lateral” or rewrite the sentence.

We agree with the reviewer and have removed the word “lateral” in the sentence.

Introduction: “several models in the literature that have been proposed”While this is technically a true statement, it is misleading. These models share features but are largely tied to the details of the biological system under study. It might be better to state that there is no generally accepted theory for the control of these sorts of movements. Indeed, the interpretations are linked to the specifics of each model system.You do not need to reject these ideas to propose your own. Indeed, I think that each of the authors that were cited have already informally considered the idea that there is a limitation for using movement for sensing related to the energetic costs. As stated in the Discussion, most of the previous work is "under-specified" which is to say that those works did not attempt to provide complete control framework as you have done. In this way, it is misleading to suggest some sort of equivalence of those previous models to your own. Your work goes beyond those previous works.The contribution of this manuscript is a formal description of information gathering via movement that both can be used to analyze biological data and drive the design of artificial systems. Its core proposition is that there is a tradeoff between energy use for movement and the statistics of information generated by that movement.Introduction paragraph one: “For example, […] Puzzling…”It is a mistake to say "For example" because this is the start of a directed argument, not just one of the various examples from the literature. What I think that I am reading here is an attempt to "soften the blow" that you deliver here, that you are directly rejecting two hypotheses from the literature – the “signal slope” and “infotaxis” ideas. I don't see these ideas are widely accepted in the literature, and so using them as contrast for your theory is not necessary. I think the manuscript would be stronger if you eliminate this unnecessary background and jump straight to paragraph two: "Here we propose…" The other ideas are reviewed sufficiently in the Discussion.

We agree with the reviewer. The Introduction is rewritten in the revision to incorporate the suggestions. We have removed the text of “For example,.…” until “Puzzling” and replaced it with an explicit statement of the core of the model. Overall, we attempt to make the new Introduction:

1) Explicitly introduces the core proposition of the EIH model—that there is a trade-off between energy cost of movement and information gain from the movement.

2) Eliminates unnecessary background (as suggested by the comment, it is covered in abundant detail in the Discussion).

Figure 1: I did not find this figure to be particularly informative. These traces do not look similar to my eyes, and I think that most people would also struggle to find exactly what compelling information is intended here. The caption states that these trajectories are "strikingly similar" but I can't help but imagine that an equally plausible caption would claim that the trajectories are "strikingly different."Perhaps you are trying to highlight the fact that there are two categories of movement, task-related which is lower frequency and sensing-related which is higher frequency. But it is difficult to extract that without some stimulus to compare against. And consider, for example, that the cockroach movement which looks like a large oscillation around a mean while the snail has small, high-frequency oscillations with a baseline that is decreasing. And what is the reader expected to see in the nautilus panel? In sum, this figure is "telegraphic" in that the intended interpretation of the figure relies on information that is not available to the reader. Although I suspect that you will want to revise this figure, it is my suggestion that you simply eliminate it. And if you can't bring yourself to eliminate it, please move it to the supplemental.

The reviewer is correct that one of our intended messages from this figure is to “highlight the fact that there are two categories of movement, task-related which is lower frequency and sensing-related which is higher frequency”. We agree with the reviewer’s points about Figure 1 and have moved the figure to the supplement (see Figure 1—figure supplement 1) to serve as an overview of the controlled sensing behavior across a wider range of animal species (including those that are not included in the main paper due to lack of data). We believe this change provides the context needed to avoid potential confusion and imply the breadth of animal species potentially concerned by the model.

On a side note – I am worried that the images are unreferenced uses of copyrighted materials from other works. More than worried, in truth. I used one of those images in one of my own efforts and received an e-mail complaint from the person who generated it. That person will certainly read this paper and I suspect be similarly annoyed. Please provide the appropriate citations/references for those images or replace them with your own materials.

We thank the reviewer for the suggestion as we had not realized this potential issue at all. In order to prevent this from happening, we have carefully cross-referenced all the included animal images and selected only those available from Wikipedia that are licensed under Creative Commons public license. Those licenses are also explicitly referenced to make sure the original authors are acknowledged for their work. (See Figure 1—figure supplement 1 and 6).

Figure 3: Please replace the trajectory example used as the icon for ergotic harvesting. It is confusing because it looks like data. I'm not sure what icon to use for the model instead, but please try something else.

We agree with the reviewer. We should also note that similar suggestion is also commented by reviewer 2. To avoid the confusion, we have therefore removed the icon of the EIH as functionally it adds no value for this figure. (See Figure 6.)

Figure 4: This is a complex figure that, rather than convincing me of the usefulness of the model, instead had me questioning it. In Figure 3, the trajectories are “eyeball-o metrically” convincing. Here in Figure 4 is an approach to quantify that perception. The challenge is that the data sets are quite different from each other, requiring different choices about how to analyze the data. These differences make the analysis appear arbitrary, as each panel is different. Further, the model does not produce output that matches the behavioral data – the data in the right most column do not match the data in the adjacent column. The main finding, that there is an increase in active movements in relation to the strength of the signal, is lost among the many details here.It is my guess that the result is likely robust to the details of the frequency window, and perhaps you can devise a simpler window or analysis that can be applied to all of the examples, and then show the main result only. Perhaps, for clarity, the Fourier plots could be moved to supplemental for deeper discussion.

We thank the reviewer to points out the potential confusion introduced by presentation of statistical results that made them visually differ. In the process of revision, we noticed that this inline comment is different from the summary comment.

After communicating with the editor asking for clarifications to make sure we have interpreted the reviewer’s final intention properly, we have hence kept the Figure 4 in the main draft (now becomes Figure 7) while revised the statistical panels to offset the mean of the strong signal’s Fourier magnitude. By making this change we hope to deemphasize the absolute difference between the behavior data and EIH model which is not what we intend to focus, but rather the fact that weak signal’s condition always have a significantly higher spectrum power (see Figure 7). We would appreciate feedback from the reviewer as to whether we have interpreted their intent correctly.

Figure S1: I was surprised to find this figure in the supplemental. I think that this should be Figure 1 of the main text as it is the conceptualization of the main contribution of the paper, which is the control model that shows how animals simultaneously manage the uncertain, non-uniform distribution of information in the environment and the costs of movement.

We agree with the reviewer and have made the following changes:

1) Figure S1 is moved to the main text as the new Figure 1.

2) We also added a new panel (panel Figure 1 A) to provide more context in helping the reader understand the information landscape and that one of the “target” is actually a distractor (with corresponding text in the Introduction defining the concept of distractor).

Figure S2: This figure covers an issue that I think will be of interest to most readers, and I also strongly suggest moving this to the main text.

We agree with the reviewer. The belief versus EID comparison plot for weakly electric fish EIH simulation found in Figure 6—figure supplement 1 is integrated to the new Figure 2. Figure 6—figure supplement 1 is still included in the Supplement to keep providing finer detail about the EIH simutions including the vertical line marking each planning epoch and annotations for fictive distractors seen in the belief, while avoid making Figure 2 too complex as an introductory figure.

Moving these two supplemental figures to the main text and eliminating the current figure 1 will focus the manuscript on its important contribution, the model.

We agree with the reviewer and have eliminated the current Figure 1 and incorporated Figure S1 and S2 into the new Figure 1 and 2.

OK – to wrap up. I strongly suggest a rewrite with a focus on the model, transforming this paper from reading like a review paper with a model added to a research paper with good scholarship. I would start your edits with the title, which should focus on the main point. I'll hazard a suggestion just to get things rolling – "Tuning movements for sensing in an uncertain world"“Sense organ control in moths to moles is a gamble on information through motion”

We appreciate the reviewer’s thoughtful comments. The suggested rewrite helps make the paper model-centric and, as reviewer has already pointed out, improves the overall presentation of the paper that ultimately contributes to better readability. Agreeing with the reviewer’s suggestions, we have carefully taken the effort to rewrite the paper with a strong focus on the new theory. This is represented in:

1) The arrangement of the updated figures that start with a direct introduction of the concept of energy-constrained proportional betting and an overview of components of its algorithmic implementation;

2) Rewritten text in the Introduction and Results section that highlights the predictions of energy-constrained proportional betting theory in the context of the fish data, followed by the expanded analysis of generalizing this theory to other animal species.

Finally, we thank the reviewer for suggesting a better title, which we have adopted but with the change of singular rather than plural for “movements”.

We hope the reviewer finds the rewrite helpful in transforming the paper into a model-centric paper with comprehensive scholarship rather than a review paper.

Also, I think that haptic touch and whisking, among the most well-studied systems in which movement is used for sensing, should be better represented in the manuscript.

We appreciate the suggestion from the reviewer and agree. We have hence included citations of active whisking papers in the Introduction to representation the lines of research. In addition to the cited works, given our limited knowledge in the area of haptic touch and whisking, we would be happy to consider any additional suggested works that the reviewer thinks we should include.

Reviewer #2:The paper by Chen et al. poses a new strategy for interpreting active sensation produced by animal movements. Established in earlier work, the expected information distribution (EID) specifies how future observations would reduce the uncertainty (entropy) of the belief that an agent (animal or robot) has developed about the location of a sensory object. As I understand it, previous work has developed infotaxis, a search algorithm to move towards the position (or other state) where there is a maximum in the EID. Here, the authors identify a problem with this earlier approach that it may allow convergence to distractors and favor locking to an offset point during tracking. The authors use a new strategy (ergodic information harvesting – EIH), posed in an earlier paper, that plans an optimal route through an EID to best reduce uncertainty in the sensory representation while balancing energetic costs. This routing is iteratively refined as the estimate of the object position, the belief, is updated. The authors suggest that this strategy avoids local convergence by encouraging exploration and establishing a tradeoff which may explain the "wiggles" and occasional large divergences animals show in the trajectories they take.Overall I find this a very compelling presentation and assessment of an interesting hypothesis. The insights that the authors have into active sensing and information acquisition are interesting and represent a clear advance in thinking. The data mining and reanalysis of published literature is clever, carefully done on the whole and seems excellent. While I have a few comments for clarification, I am enthusiastic about this paper and think it will make a significant contribution to the literature.The EIH algorithm has been posed earlier, but the authors take a very significant step forward in providing rigorous testing of this hypothesis in the context of single target tracking in several different animals. I especially like the breadth of systems. The prior algorithm development should be more clearly cited in the Introduction.There is potential concern that the given experiments do not fully reject all alternative models. I don't think this is a deep problem. The authors acknowledge that a gain adjusting infotaxis model might make some of the same predictions at the EIH algorithm and discuss several alternatives. The contrast with fixed gain infotaxis is clear and the alternatives suggest future work to distinguish the "right" modification to simple EID based strategies.The EIH algorithm seems to make predictions that deviate from infotaxis in two ways. One is when there are multiple targets and one is when there is uncertainty about a target such that a "false" target might be perceived. The authors argue that the latter is what they test and that the former while interesting is beyond the data that they have. I do find the repeated use of the two target EIH simulation a bit distracting when incorporated into the figures with actual animal data (Figure 3) because those experiments don't reflect that part of hypothesis and those specific predictions. This is especially confusing when comparing to tracking data with a moving target. I understand that the second target could be a "false" target rather than a physical multiple target, but it is difficult to gain an intuition that connects the animal data and this example where the second target is a fixed point in space.In addition to clarifying the figure to avoid confusion, I think the authors should make the distinction about the two different EIH conditions (uncertain single targets, and dual targets) more clear in the Introduction. They can then focus on the former with the animal data, confidently simulate the second, and suggest experiments for the latter as they do in the excellent treatment of this in the Discussion. The way the manuscript is currently framed makes the distractor experiments sound very appealing because they would seem to be strong at distinguishing between the various alternative explanations in the Discussion.

We thank the reviewer for the thorough review and the constructive feedback. In the revision, we have carefully considered their concerns and suggestions, and have rewritten most of the Introduction and Results of the paper in addition to updating and rearranging most of the plots in the main text.

We sincerely appreciate the reviewer for the recognition of the “significant step forward” that we aim to provide with this work. We also relate to the concern that the reviewer is raising on both that 1) the included analysis cannot fully reject all the alternative models and 2) the EIH algorithm along with the key concept of fictive distractors are not clearly defined in the initial submission. While the former goes beyond this work, we have nonetheless updated the Introduction to adjust the language when introducing the EIH model to make it no longer sounds like we are rejecting all alternative models (similar feedback is also raised by reviewer 1). We also addressed the latter by:

1) Remaking the figures and removing the distracting EIH visualization where it shows the static two target tracking in 2D example (see Figure 6);

2) Replaces the less intuitive introduction Figure 1 with an enhanced version of what used to be Supplementary Figure S1 (see the new Figure 1) and introduces the concept of fictive distractors early in the paper (see Figure 2);

3) Added additional text that introduces the concept of fictive distractors and real distractors in the Introduction.

The reviewer also made some specific suggestions in their summary comment. We

have included our responses to these as well as the original feedback in the individual comments below.

The prior algorithm development should be more clearly cited in the Introduction.

We agree with the reviewer and have made sure to explicitly cite the prior algorithm development work in the introduction when EIH is first introduced. However, we have also slightly reframed the presentation to forward the point that this is a new theory for the origin of sensing-related movements. We do this by distinguishing between the theory of energy-constrained proportional betting (for sensing related motions) which is not present in the earlier work, from its implementation—ergodic information harvesting—an algorithm which in some significant measure was present in the earlier work in the context of controlling a robot. In this way we parallel the infotaxis study, which similarly uses a quantitative framework that was already well established toward a new theory for the movement of sensors.

I do find the repeated use of the two target EIH simulation a bit distracting when incorporated into the figures with actual animal data (Figure 3) because those experiments don’t reflect that part of hypothesis and those specific predictions. This is especially confusing when comparing to tracking data with a moving target. I understand that the second target could be a ”false” target rather than a physical multiple target, but it is difficult to gain an intuition that connects the animal data and this example where the second target is a fixed point in space.

We agree with the reviewer. In response to this feedback (and together with similar comment from Reviewer 1 about the icon for EIH is confusing as it resembles data where it is not), we have updated the figure (see Figure 6) to remove the EIH icon that invites confusion.

In addition to clarifying the figure to avoid confusion, I think the authors should make the distinction about the two different EIH conditions (uncertain single targets, and dual targets) more clear in the Introduction.We agree with the reviewer and have added a new figure (see Figure 2G) and text (see Introduction section) to clearly introduce the concept of distractor and disambiguate between fictive and real distractors.

[Editors' note: further revisions were suggested prior to acceptance, as described below.]

Reviewer #1:1) I feel that sections of the Introduction could be shortened or omitted. In the Introduction, most of the paragraph that starts with "An important quantity for implementing energy-constrained proportional betting…" could be deleted. I understand that this example is included to help the reader, but I felt like it didn't add much. It doesn't help because 1) the example of the fish seems clear enough, and 2) adding yet another example, finding a WiFi router, doesn't add much intuition. Indeed, that and the following two paragraphs would, I believe, be clearer if they were shortened and more focussed.

We have combined reviewer 1’s comment here with reviewer 2’s, and changed the example to use the moth. We feel that this communicates the introductory material well.

2) In a similar vein, I was surprised by Video 1, which rather than showing data from the manuscript, is a form of tutorial using other (sometimes unrelated) systems and tasks. I think your readers would appreciate videos of the animals in the manuscript – the primary data – perhaps side-by-side with the simulations. As most readers are unlikely to be directly familiar with these behaviors, that would be a contribution. Further, you could assemble video footage for each of the systems that you mention in the paper, something I would like to see! Also, the video is rather uneven in its production, particularly with regards to matching the narration to the video being shown. Please either replace them with data footage, or if you decide to keep the current strategy, spend some more time editing the video content to better match the narration.Video 2 is a tutorial of the method, which I found (although unusual outside of JoVE) to be useful.

We have changed Video 1 to only contain segments of the analyzed behavior, and kept Video 2 as is.

3) Consider adding a paragraph on "Optimal Foraging Theory" in the Discussion. As you know, these models explore the relations between resource distribution, cost of locomotion, and costs of predation (among other factors). These models were particularly popular in the 1990s, involving an approach that is at least parallel to this paper – examining animal behavior in relation to the performance of optimized models. I am not sufficiently familiar with this literature to suggest a paper that generated animal trajectories similar to what was done here. I do recall the Stephens and Krebs book (1986), and I remember reading a paper that had a comparison of locomotor strategies for foraging (https://dx.doi.org/10.1073%2Fpnas.98.3.1089).

This was a very good suggestion and we have added a discussion of optimal foraging theory to the Discussion.

4) I didn't mention this in the previous review, but I thought that it might be interesting to add a few sentences in the Discussion about the differences in perspective between your approach and perspective versus our perspective in the Stamper, 2012 active sensing paper. The data on fish tracking are remarkably similar (coincidentally Figure 3B in both your manuscript and our 2012 paper). Our attention was turned towards the brain and controller – proximate mechanisms – whereas the focus here is on “evolutionary” impacts – ultimate mechanisms. This difference in perspective has consequences that may be interesting to discuss. As you know, we did not build a model in the way you did here, but the structure (and implications) of that model would have been quite different. I leave it to you to decide what might be useful (if anything) to add to the Discussion.

We appreciate the point here, but given the very long Discussion we think it would be better to keep the Discussion as is (with the added section on optimal foraging addressed above).

Reviewer #2:Overall I think the authors have done an admirable job responding to all of the review comments. I think this paper will make a nice and somewhat provocative addition to the literature. Given the extent of the changes I did have one follow-up comment. The authors have made many changes to the Introduction and I appreciate their efforts. However, I think the change in Figure 1 has gone a little too far in the opposite direction. While I appreciate the change to Figure 1 and the attention to minimizing the discussion of the two target case, it is now quite confusing to have this be the only figure in the Introduction. The Introduction now lacks a figure with strong biological ties especially because the Introduction is framed around an uncertain single target search or tracking problem, but the first figure now is only the two target case. Figure 1—figure supplement 2 is actually discussed more in the Introduction. That figure alone is also not sufficient because of the increased emphasis on the fish work. I really like the two target tracking simulation in Figure 1 and it should be in the paper, but it is confusing as the only motivation figure in the Introduction. Perhaps the easiest solution is to add Figure 1—figure supplement 2 or some of the examples from Figure 1—figure supplement 1 back into main Figure 1 keeping in mind the other reviewer's concern that some of the data in Figure 1—figure supplement 1 (originally Figure 1) were a bit underspecified. Alternatively a more schematic or simulated example of the tracking case could be included and discussed but this seems like more work.

We have modified Figure 1 following reviewer 2’s helpful pointers.